# Spatiotemporal characteristics and influencing factor analysis of universities' technology transfer level in China: The perspective of innovation ecosystems

**Haining Fang[1], Jinmei Wang[2]\*, Qing Yang[1,2], Xingxing Liu[2], Lanjuan Cao[2,3]**

**1** School of Management, Wuhan University of Technology, Wuhan, China, **2** School of Safety Science and Emergency Management, Wuhan University of Technology, Wuhan, China, **3** School of Mechatronics and Energy Engineering, NingboTech University, Ningbo, China

\* wangjinmei@whut.edu.cn

## Abstract

Universities are important parts of innovation ecosystems, and university technology transfer (UTT), which aims for the sustainable commercialization of sci-tech achievements, is closely related to other actors in the ecosystem. Based on the panel data of 31 provinces in mainland China, this paper empirically analyzes the spatiotemporal distribution characteristics of UTT levels from 2011 to 2019 and estimates the influencing factors using the spatial Durbin model (SDM) with an economic spatial weighting matrix from the perspective of innovation ecosystems. The results are presented as follows: (1) Although the overall level of UTT in China is low, it shows an upward trend in most provinces. In addition, the interprovincial gap is obvious, forming a ladder distribution of UTT levels increasing from west to east. (2) There is a significant spatial autocorrelation between UTT levels in the provinces. (3) Industry, economy, and informatization play significant roles in promoting UTT, while financial institutes and openness have significant inhibitory effects. The economy has a significant spatial spillover effect on UTT, while government, industry and informatization have a significant inhibitory effect on UTT in neighboring regions. (4) The direct and indirect effects of influencing factors in the Eastern Region and other regions show significant spatial heterogeneity.

## 1. Introduction

Universities are well-defined centers of teaching, research, and collaboration with industry for innovation [1], which play a key role in the national innovation system [2]. As the birthplace of advanced scientific knowledge and incubators of cutting-edge science and technology [3], universities influence a country's innovative potential and can thereby act as driving forces for economic development, technological performance, and competitiveness. Meanwhile, universities take an active role in commercializing knowledge through spin-offs, patents, and licensing [4], and university technology transfer (UTT) is the process of transferring sci-tech

**Funding:** This research was funded by the 2020 Social Science Foundation Key Project of Hubei Province (New Think Tank Project) (No. BSKZD2020002), 2020 Universities Practical Education Project of Hubei Province (No. 2020SJJPE3004) and Hubei to create a highland of science and technology innovation in the optical and electronic information industry status and countermeasures research (No.2022EDA005). The funders had no role in study design, data collection and analysis, decision to publish, or preparation of the manuscript.

**Competing interests:** The authors have declared that no competing interests exist.

achievements into marketable products and services [5,6]. UTT is of vital importance in many respects; for example, it represents a source of funding for university research and innovation for businesses and, most notably, a source of economic development for policymakers [7]. As the major cooperation channel between universities and industry, it promotes the transfer of knowledge and technology from universities to industry and society. Therefore, in the era of a knowledge-based economy, governments have been seeking to boost the UTT process and promote its application [8]. A prominent and well-researched example is the Bayh-Dole Act formulated in the United States in 1980, which addressed the rule predicament by facilitating universities to transfer their achievements independently [9,10].

Over the past 20 years, universities have made steady progress in their efforts to facilitate the technology transfer process by working with industry [7], but the UTT level is still remarkably low. Approximately 20% of disclosed university innovations are commercialized yearly in the US, and the remaining 80% are unlicensed [11]. The annual income from licensed university inventions was 1.7% of total research expenditure in 1995 and 2.9% in 2004 [12]. In China, the government provides a large amount of yearly financial support to universities conducting scientific research. However, the patents brought in from UTT, in stark contrast to those in the US, are rather low. According to the statistics of the "China Intellectual Property Survey Report 2019" by the China National Intellectual Property Administration, the effective patent enforcement rate of Chinese universities in 2019 was 13.8%, which was far lower than the national average of 55.4%. In addition, the effective patent industrialization rate of universities was only 3.7%, which was also far lower than the national average of 38.6%. A large number of sci-tech achievements did not transfer to productivity or promote economic and social development. Over the past three decades, China has made unprecedentedly large investments in university research, and Chinese universities are now becoming major contributors to the global scientific community [13]. However, it can be seen from the data that the proportion of UTT is still at a low level, which does not match economic development. As the second largest university research system in the world, a better understanding of the complex ways of Chinese UTT is critical for scholars and policymakers globally. Moreover, improving the Chinese UTT level can enhance innovation vitality and further promote economic development. Therefore, studies on the influencing factors of UTT are of great significance.

Transforming the idle sci-tech achievements of universities into social productivity, which involves the government, universities, enterprises, sci-tech intermediaries, and financial institutions and is influenced by factors such as policy, mechanism, institution, capital, and manpower, is a complex and consistent innovation process. As a heterogeneous post developing country, China faces a host of problems, such as complex industrial characteristics, significant differences in regional economic development, and uneven distribution of regional science and technology resources [14]. Some scholars have noted the heterogeneous impact of the subnational environment [15], social trust [16], and promarket reform [17] on UTT. Hou et al. [18]. discover that the moderating role of intermediary organizations in academia-industry cooperation and industrial innovation varies in inland and coastal areas. The sustainable development and transformation of knowledge and innovation should be considered a dynamic and systematic coevolutionary process [19]. However, traditional ordinary least squares (OLS) methods ignore the effect of spatial overflow. There are few studies systematically analyzing the heterogeneous impact of regional innovation ecosystems on UTT. Compared with the literature, this paper (1) establishes a comprehensive evaluation index system of the UTT level based on the evaluation index screening and weighting method with the information contribution rate and (2) analyzes the heterogeneous impact of innovation actors on UTT from the perspective of an innovation ecosystem.

The paper is organized as follows. Section 2 reviews the literature on UTT and proposes hypotheses. Section 3 introduces the research methods and data sources. Section 4 presents the spatiotemporal characteristics of the UTT level. Section 5 is devoted to analyzing the spatial effect of the influencing factors. The final section discusses the results and policy implications.

## 2. Literature review and hypotheses

### 2.1 UTT level

The evaluation of UTT capability is an important research direction of UTT, most of which measures efficiency. Data envelopment analysis (DEA) and stochastic frontier analysis (SFA) are the most common methods. Some scholars divide the process of UTT into two [20,21] or three stages [22] to measure its efficiency, which contains basic research, application research and experimental development. Without dividing UTT into stages, some scholars select different input–output indicators [23,24]. Some scholars use SFA for measurement, which can test hypotheses and build confidence intervals after estimation to differentiate inefficiency and noise [25]. However, it needs to set the corresponding function in advance. In contrast, DEA is a nonparametric model without functions to be defined, and the effects of the form might not be mixed with those of inefficiency [26]. However, the efficiency value measured by the DEA method is highly sensitive to input–output indices and cannot measure scales. In the case of few input–output indices and incomplete data statistics in China, the efficiency value can only slightly reflect the true UTT capability. Apart from efficiency measurement, another way to evaluate UTT capability is to establish a comprehensive index evaluation system [27,28], which is relatively comprehensive and can reflect the actual level of UTT to a certain extent.

These studies have focused on the measurement of UTT levels, so it is necessary to explore the performance patterns of UTT in different provinces through spatial statistics and analysis and to conduct predictive simulation studies on the aggregation characteristics and association patterns of UTT in space.

The core of UTT is technology and knowledge spillover, which depends on the interaction between industries and universities [29]. Improving the level of UTT can enhance the interaction between industries and universities, but the development of UTT is very uneven among different regions of China. It is urgent to allocate scientific and rational research resources, narrow the gap between universities' research levels, and improve resource allocation levels and UTT levels by spatial autocorrelation analysis. Furthermore, geographic proximity increases the possibility of collaboration between universities and industries [30] since the exchanged knowledge is cumulative, localized, and tacit in nature. In addition, geographic proximity can also strengthen other forms of proximity, such as cognitive, organizational, and technological closeness [31], which are essential for UTT. Thus, the paper proposes the following hypothesis:

H1: The provincial UTT level shows spatial autocorrelation in China.

### 2.2 The influencing factors of UTT level

The analysis of the UTT level in China is essentially the spatial differentiation of the knowledge output of universities in different regions and the spatial imbalance of research activities in universities in different regions. Specifically, the level of UTT in China is influenced by many factors. On the one hand, some of the influencing factors are internal to the university, such as the lack of well-trained staff and inventions processing capacity [12], insufficient rewards for faculty involvement in UTT [32], university's academic eminence [33], medical schools [34] and science parks [35]. Sun [36] used the DEMATEL information analysis method to discuss the key

UTT internal factors, with results showing that the effect of different factors on UTT widely varied. Institutions and infrastructure are supportive of increasing university patents [37,38]. Lafuente et al. [39] analyzed the productivity of Spanish technology transfer offices (TTOs) during 2006–2011 by computing total factor productivity models rooted in nonparametric techniques (Malmquist index), and the results confirmed that changes in the configuration of the TTO outcome portfolio affect UTT. Wonglimpiyarat et al. [40] compared the operation of university business incubators (UBIs) and technology incubators to understand different technology transfer strategies and argued that incubation programs were one of the main policy mechanisms to support innovation, connect the university and industry sectors, provide interactive links and effectively promote university research. Erika et al. [41] designed a questionnaire for the willingness of researchers to participate in UTT and concluded that the active participation of researchers had a crucial role in promoting UTT. Li et al. [42] built a technology acceptance model and detected that the inventor's technology service acts as a mediator in the relationship between university technology transfer sustainability and influencing factors. In addition, tacit knowledge transfer will directly affect university technology transfer.

On the other hand, external factors can also influence UTT levels. The innovation ecosystem provides a suitable framework for the analysis of UTT. A vigorous innovation ecosystem will provide all the actors with an innovative environment of "tropical rainforests". In an innovation ecosystem, actors are interdependent against each other and inseparable from each other's behavior [43,44]. The innovative activities of universities and UTT are very closely related to other actors in the innovation ecosystem. The influence of other actors on UTT is as follows: (i) Government: In an innovation ecosystem, government investment can guide innovation through policies to affect UTT. The logic of government policies, represented by the Bayh-Dole Act, is to stimulate universities to support and build infrastructure for technology commercialization [45]. However, the study found that some policies have met their expectations, while other policies have not [46]. (ii) Industry: A high industrial development level means a good industrial cluster formed in the area, and the demand for technological innovation is stronger. It provides a rich market for UTT [47]. The role of the government [48] and the high-tech industry [49] are both key parts of UTT activities. (iii) Financial Institute: Experience showed capital investment in the stage of industrialization of sci-tech products requires 10 times as much as technology transfer, and 100 times as much as sci-tech research and development. The lack of funding sources constitutes a major barrier to effective UTT [50]. A better level of financial development can provide good support for investment and financing, information disclosure, risk diversification and price discovery in the process of UTT [51]. (iv) Technology Intermediary: On the one hand, technology intermediaries can improve cognitive and organizational dimensions and reduce social and geographical distance [52]. On the other hand, it can help universities and industries develop trust, enable partner identification and thereby enhance successful collaboration [53]. Hence, the connection between universities and industries can be effectively promoted. There is evidence to show that intermediary organizations are crucial in improving universities' patent performance [52,54].

Based on the above literature, we summarize the innovation ecosystem around UTT, as shown in Fig 1. The framework of the innovation ecosystem extends the study of UTT from university-industry cooperation (UIC) [55] to all actors' coordination. In the innovation ecosystem, technology, funds, policy, investment and service act as the energy flow. Outside the innovation ecosystem, economy, informatization and openness can also affect UTT. In addition, we propose the following hypothesis:

H2: The factors in the innovation ecosystem have a positive impact on UTT.

H3: The influence of the factors on the UTT level is spatially heterogeneous in China.

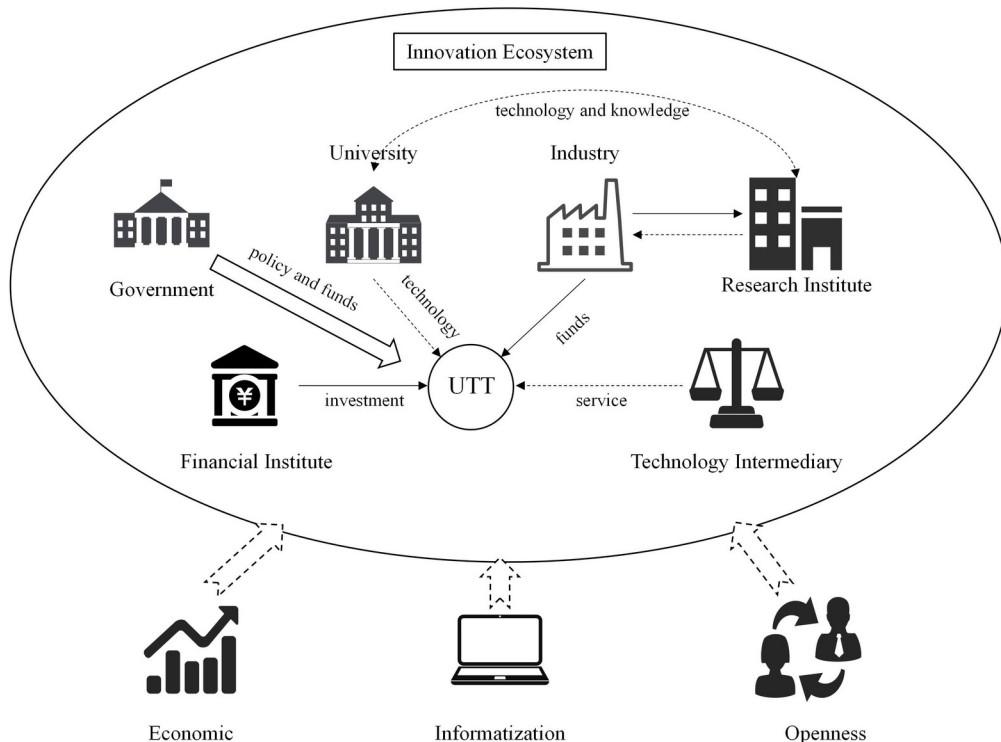

**Fig 1. The impact of innovation ecosystem on UTT.**

## 3. Methodology and data sources

The methodology model of this article is shown in Fig 2. First, the method of UTT level evaluation is selected. Second, Spearman rank correlation and Moran's I are used to analyze the spatiotemporal characteristics of the UTT level. Third, after determining the existence of spatial autocorrelation of the UTT level using Moran's I, spatial econometric analysis of the influencing factors is carried out. It includes influencing factor selection and spatial effect analysis.

### 3.1 Evaluation of UTT level

This study adopts the evaluation index screening and weighting method based on the information contribution rate, which is a comprehensive index evaluation method.

The evaluation index screening and weighting method based on the information contribution rate effectively retains the indices with strong information interpretation ability and reduces the overall information overlap level of the index set. The steps for screening indicators and determining weights include (1) calculating the eigenvalues of the index correlation coefficient matrix, (2) determining the key factors to be retained, (3) calculating the factor loading matrix, (4) calculating the information contribution rate of the indices, (5) calculating the cumulative information contribution rate, (6) screening indicators with a high information contribution rate, (7) checking the necessity of eliminating information overlap indicators, (8) eliminating indicators with a high degree of information overlap, and (9) determining the weight of the evaluation indices. The detailed calculation process can be found in the related reference [56], and the results are shown in Table 1. To reflect the comparability of data between provinces, all indices are used per capita data. In addition, the "Actual income in the

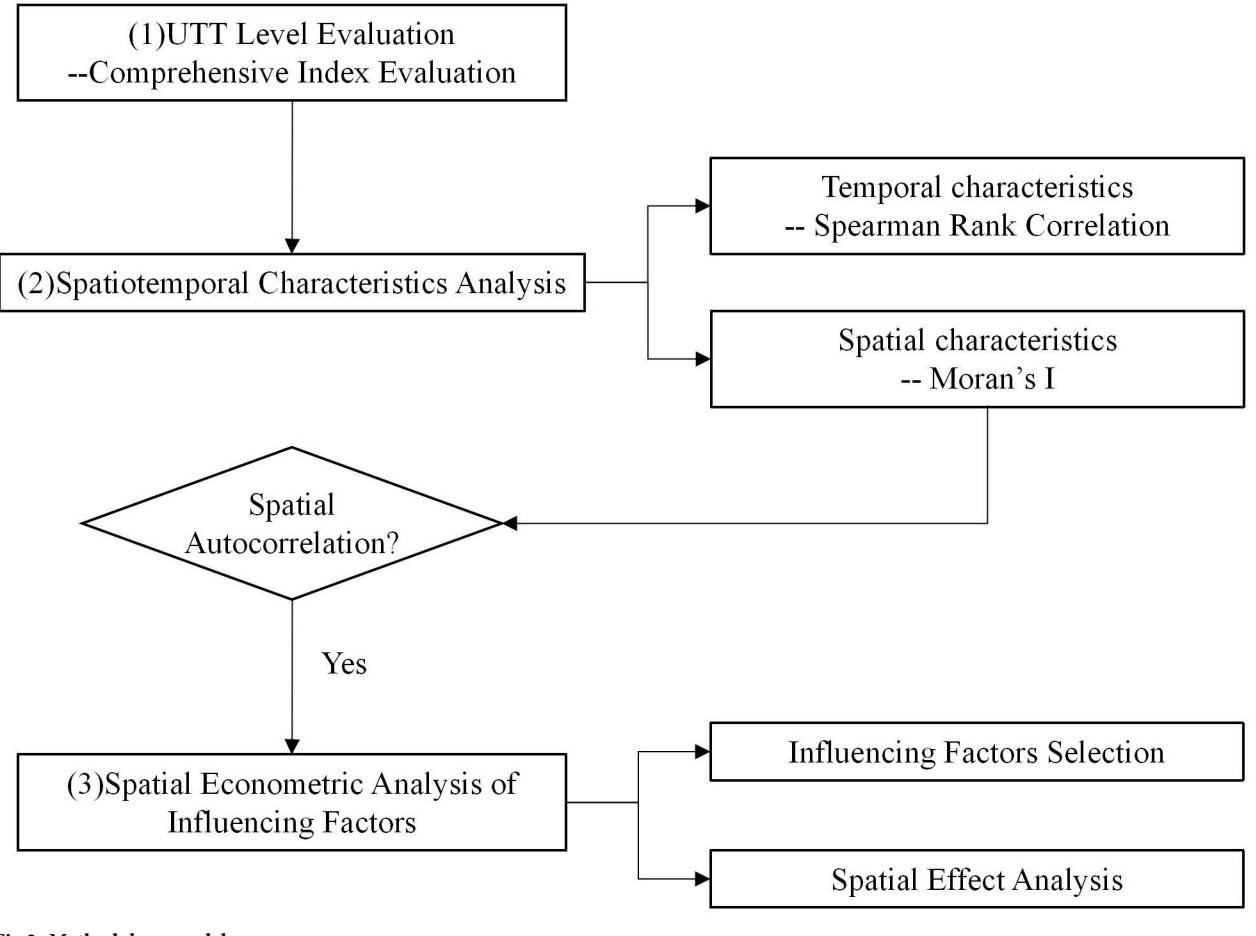

**Fig 2. Methodology model.**

year of UTT/Thousand yuan" index is measured at constant prices, and the original data have been processed.

According to Table 1, the evaluation formula for the UTT level is as follows:

$$U = \sum_{r=1}^{p} u_r \omega_r \qquad (1)$$

$u$ is the standardized value of the UTT level evaluation index, $\omega$ is the weight corresponding to the index, and $p$ is the number of indices.

**Table 1. Evaluation index system of UTT level after screening.**

| Screened indices | Weight |
|---|---|
| (i) Published sci-tech work | 0.3418 |
| (ii) Actual income in the year of UTT/thousand yuan | 0.2597 |
| (iii) Number of signed contracts for UTT | 0.2108 |
| (iv) Number of patents granted | 0.1877 |

## 3.2 Spatiotemporal characteristics analysis

**(1) Temporal characteristics.**   To better show the temporal characteristics of the UTT level of each province, the Spearman rank correlation coefficient [57,58] method is used to analyze the changing trend of the level from 2011 to 2019. The calculation formula of the Spearman rank correlation coefficient is as follows:

$$R = 1 - \left[ \frac{6 \cdot \sum_{t=1}^{m}(X_t - Y_t)^2}{M^3 - M} \right] \tag{2}$$

$R$ is the Spearman rank correlation coefficient, $t$ is the year, $X_t$ is the serial number from 2011 to 2019 in order of efficiency from small to large, $Y_t$ is the serial number by time, and $M$ is the number of samples. The absolute value of $R$ is compared with the critical value $W_P$ in the rank correlation coefficient statistics table. If $|R| \leq W_P$, it indicates that the changing trend is significant. When $R$ is a positive value, the UTT level shows an upward trend, and when $R$ is a negative value and shows a downward trend.

**(2) Spatial characteristics.**   Before conducting spatial econometric analysis, whether the data are spatially correlated should be verified. Spatial autocorrelation can be understood as nearby areas having similar variable values. If the high values are clustered together and the low values are clustered together, it is called positive spatial autocorrelation; conversely, if the high value is adjacent to the low value, it is called negative spatial autocorrelation. If the high and low values are completely randomly distributed, there is no spatial autocorrelation [59,60]. The common method for evaluating spatial autocorrelation is Moran's I:

$$I = \frac{\sum_{i=1}^{n}\sum_{j=1}^{n} w_{ij}(x_i - \bar{x})\left(x_j - \bar{x}\right)}{S^2 \sum_{i=1}^{n}\sum_{j=1}^{n} w_{ij}} \tag{3}$$

$S^2 = \frac{1}{n}\sum_{i=1}^{n}(x_i - \bar{x})^2$ is the sample variance, $\bar{x} = \frac{1}{n}\sum_{i=1}^{n} x_i$, $x_i$ and $x_j$ represent the UTT levels of provinces $i$ and $j$, respectively $n$ is the number of provinces, and $w_{ij}$ is the element of the spatial weight coefficient matrix $W$. The space of Moran's I is $[-1,1]$. Moran's I $> 0$ represents positive spatial autocorrelation, while Moran's I $< 0$ indicates negative spatial autocorrelation. When Moran's I is close to 0, the space is in random mode, and there is no spatial autocorrelation. If the observed value and its spatial lag are drawn as a scatter plot, it is called Moran's I scatter plot. Moran's I is the slope of the regression line of the scatter plot, which can further explore the spatial agglomeration mode of the research object.

There are two commonly used setting methods for the spatial weighting matrix. The conventional method is based on the geographical proximity of the regions [61]. If provinces $i$ and $j$ are adjacent, the weight is 1; otherwise, the weight is 0, which forms an adjacency weight matrix. The second method is based on geographic distance [62], of which the negative exponential decay form of geographic distance is commonly used to construct the matrix. The spatial weighting matrix established through geographic space can reflect the geographical proximity between regions, but it does not sufficiently reflect the interaction of economic bases across regions. Although they are adjacent regions, an inconsistent degree of economic development causes an inconsistent degree of closeness. It has an impact on the spatial effects of various analysis objects and needs to be distinguished. The difference in GDP per capita between regions is used as an indicator to measure the "economic distance" between regions [63]. The economic spatial weighting matrix $W^*$ not only considers the correlation between two provinces in economic relations but also avoids the estimation problem caused by the

separate use of the geographical distance weight matrix. The economic spatial weighting matrix is:

$$W^* = W \times E \qquad (4)$$

The calculation method of the spatial weighting matrix $W$ is:

$$w_{ij} = \begin{cases} 0 & i = j \\ \dfrac{1}{d_{ij}} & i \neq j \end{cases} \qquad (5)$$

$d_{ij}$ represents the distance between two provinces. It typically adopts the distance between provincial capital cities for convenience. The main diagonal elements of matrix E are all 0, and the $(i, j)$ elements of the nonmain diagonal are:

$$e_{ij} = \frac{1}{|\bar{Y}_i - \bar{Y}_j|}, \, (\, i \neq j) \qquad (6)$$

$\bar{Y}_i$ is the average GDP per capita of province i during the sample period, using the starting year of the sample as the base period and deflating according to constant prices.

## 3.3 Spatial econometric analysis of influencing factors

When there is a significant spatial correlation between the explanatory variable and the explained variables of the UTT level, the spatial panel model should be used for estimation. Generally, the spatial autoregression model ($SAR$), the spatial error model ($SEM$) and the spatial Durbin model ($SDM$) should be considered.

$SAR$ of the UTT level (expressed in utt):

$$utt_{it} = c + \rho W_{ij} utt_{it} + \alpha_k controls_{kit} + \varepsilon_{it} \qquad (7)$$

$i$ and $t$ represent province and year, respectively, $c$ is a constant term, $\rho W_{ij} utt_{it}$ is a spatial lag variable, $\alpha_k$ is the regression coefficient of each influencing factor (controls), and $\varepsilon_{it}$ is a random disturbance term.

The calculation formula of $SEM$ is:

$$utt_{it} = c + \alpha_k controls_{kit} + \varepsilon_{it}$$

$$\varepsilon_{it} = \lambda W_{ij} \varepsilon_{it} + \mu_{it} \qquad (8)$$

$\lambda$ is the spatial error coefficient, and $\mu_{it}$ is the random disturbance term.

The calculation formula of $SDM$ is:

$$utt_{it} = c + \rho W_{ij} utt_{i,t} + \alpha_k controls_{kit} + \beta_k W_{ij} controls_{kit} + \varepsilon_{it} \qquad (9)$$

Which model to choose for estimation depends on the spatial state matrix and data using the LM test, robust LM test and Wald test.

## 3.4 Data sources

The selection of proxy variables for influencing factors is shown in Table 2.

We select 31 provinces in Mainland China (due to the limitation of data collection, excluding Hong Kong, Macao and Taiwan regions) for empirical analysis of UTT-related data from 2010 to 2019. Considering the lag of UTT output, the independent variables are processed

**Table 2. Proxy variables for influencing factors.**

| Influence Factor | Proxy Variable |
|---|---|
| UTT Level (UTT) | Results by the evaluation index screening and weighting method based on information contribution rate |
| Government (Gov) | R&D investment intensity |
| Industry (Ind) | The operating income of high-tech industry per 10,000 people (10, 000 yuan) |
| Financial Institute (Fin) | The ratio of balance of deposits to loans of financial institutions at the end of the year to GDP |
| Technology Intermediary (Int) | Technology market turnover per capita (10, 000 yuan) |
| Economic (Eco) | Per capita GDP (10, 000 yuan) |
| Informatization (Inf) | The number of internet interfaces per 10,000 people |
| Openness (Open) | The investment amount of foreign-invested enterprises per 10,000 people (10, 000 dollars) |

with a lag of 1 year. A total of 9 years of panel data for 31 provinces is obtained. Data sources include "China Statistical Yearbook", "China Science and Technology Statistical Yearbook", "China Financial Statistics Yearbook", "Compilation of Statistics on Science and Technology of Higher Education Institutions", etc.

The descriptive statistical results of each variable are shown in Table 3.

## 4. Spatiotemporal characteristics

### 4.1 Provincial spatiotemporal characteristics analysis of UTT level

Table 4 shows the average UTT level of each province from 2011 to 2019. The national UTT level is generally low, and the gap between provinces is relatively obvious. Beijing has the highest level of UTT, with an average of 0.795, which is much higher than the second highest level in the province of Shanghai. This shows that Beijing concentrates all the rich resources that are necessary for UTT and that the policy effect is relatively outstanding. Shanghai, Jiangsu, Shaanxi, Tianjin, and Chongqing are also at the forefront.

Fig 3 shows the changing trend of the UTT level in China and four major regions. This figure provides the trends of UTT developmental in China, with each region showing a slow upward trend. From a regional perspective, the UTT level presents a tiered distribution that is Eastern Region>Northeast Region>Middle Region>Western Region.

Specific to each province, the Spearman rank correlation coefficient is shown in Fig 4. Nine years of data means N = 9, and the critical value of Spearman rank correlation coefficient $W_P$ = 0.600($\alpha$ = 0.05). The UTT levels of most provinces (27) show a clear growth trend, and the

**Table 3. Descriptive statistical results.**

| Variable | Samples | Average | Standard Deviation | Minimum | Maximum |
|---|---|---|---|---|---|
| UTT | 279 | 0.143 | 0.151 | 0.000 | 0.852 |
| Gov | 279 | 0.020 | 0.014 | 0.003 | 0.066 |
| Ind | 279 | 0.752 | 0.936 | 0.008 | 4.185 |
| Fin | 279 | 3.455 | 1.234 | 1.688 | 7.565 |
| Int | 279 | 0.095 | 0.281 | 0.000 | 2.302 |
| Eco | 279 | 4.991 | 2.462 | 1.312 | 14.021 |
| Inf | 279 | 0.311 | 0.193 | 0.059 | 1.041 |
| Open | 279 | 0.353 | 0.556 | 0.012 | 3.651 |

**Table 4. Average UTT level of each province.**

| Province | Average | Rank | Province | Average | Rank | Province | Average | Rank |
|---|---|---|---|---|---|---|---|---|
| Beijing | 0.795 | 1 | Jilin | 0.134 | 12 | Henan | 0.064 | 23 |
| Shanghai | 0.386 | 2 | Fujian | 0.134 | 13 | Jiangxi | 0.062 | 24 |
| Jiangsu | 0.311 | 3 | Hunan | 0.11 | 14 | Ningxia | 0.052 | 25 |
| Shannxi | 0.272 | 4 | Sichuan | 0.103 | 15 | Yunnan | 0.046 | 26 |
| Tianjin | 0.211 | 5 | Henan | 0.09 | 16 | Guizhou | 0.044 | 27 |
| Chongqing | 0.216 | 6 | Shandong | 0.087 | 17 | Guangxi | 0.042 | 28 |
| Liaoning | 0.183 | 7 | Shanxi | 0.082 | 18 | Qinghai | 0.032 | 29 |
| Zhejiang | 0.179 | 8 | Gansu | 0.082 | 19 | Xinjiang | 0.029 | 30 |
| Heilongjiang | 0.167 | 9 | Guangdong | 0.077 | 20 | Xizang | 0.01 | 31 |
| Hubei | 0.157 | 10 | Hebei | 0.067 | 21 | | | |
| Anhui | 0.14 | 11 | Inner Mongolia | 0.069 | 22 | | | |

growth trends of Beijing, Liaoning, Fujian, Qinghai, Xinjiang, and Xizang are not significant. The provinces of Shanghai, Tianjin, Heilongjiang and Anhui show a downward trend. Among them, the UTT level in Anhui shows a significant downward trend. From the raw data, the reason for the obvious decline in Anhui is that the actual income of UTT in 2013 and 2014 doubled but returned to the previous level after 2015.

## 4.2 Spatial autocorrelation analysis of UTT level

Table 5 shows the Moran's I of the UTT level in China based on the spatial weighting matrix and the economic spatial weighting matrix. Under the spatial weighting matrix, although the

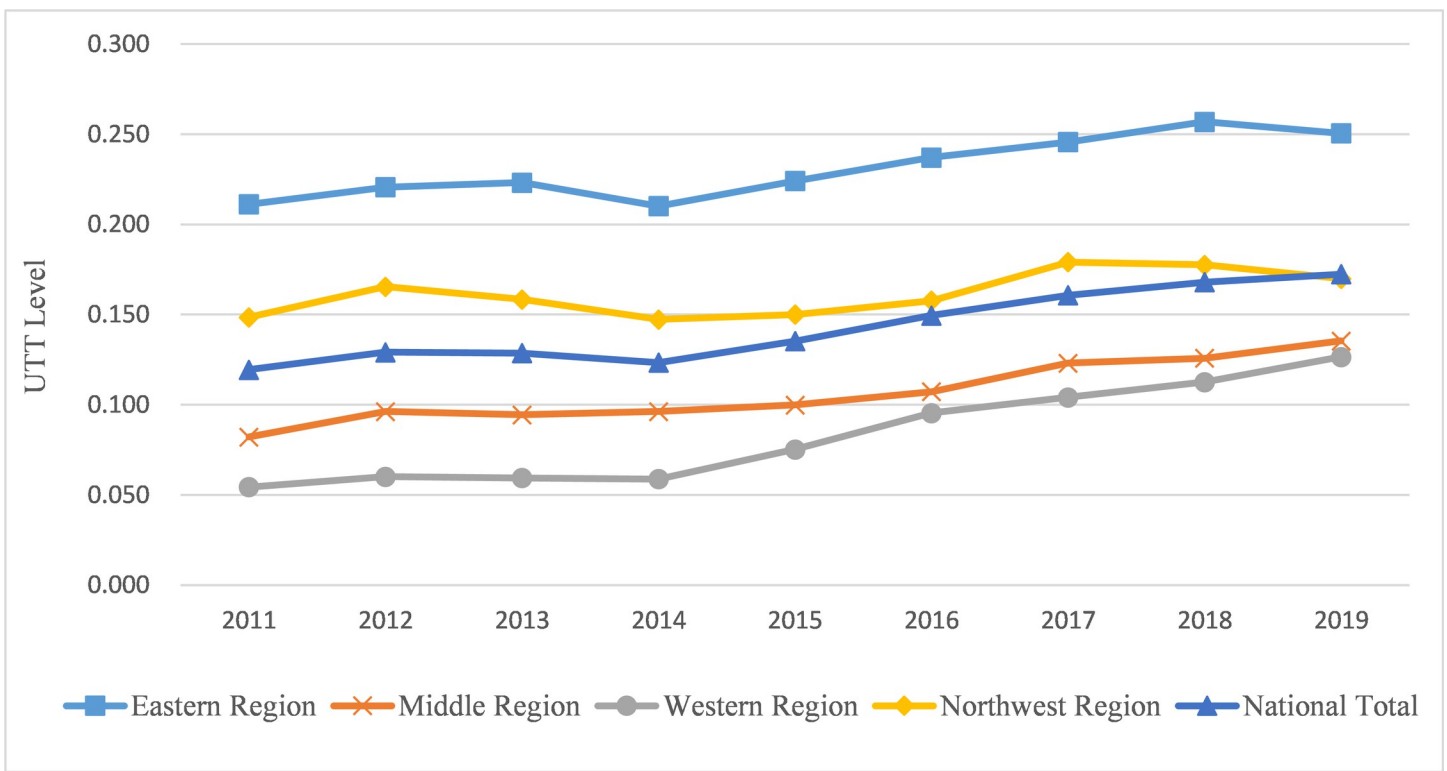

**Fig 3. UTT level in China and four major regions.**

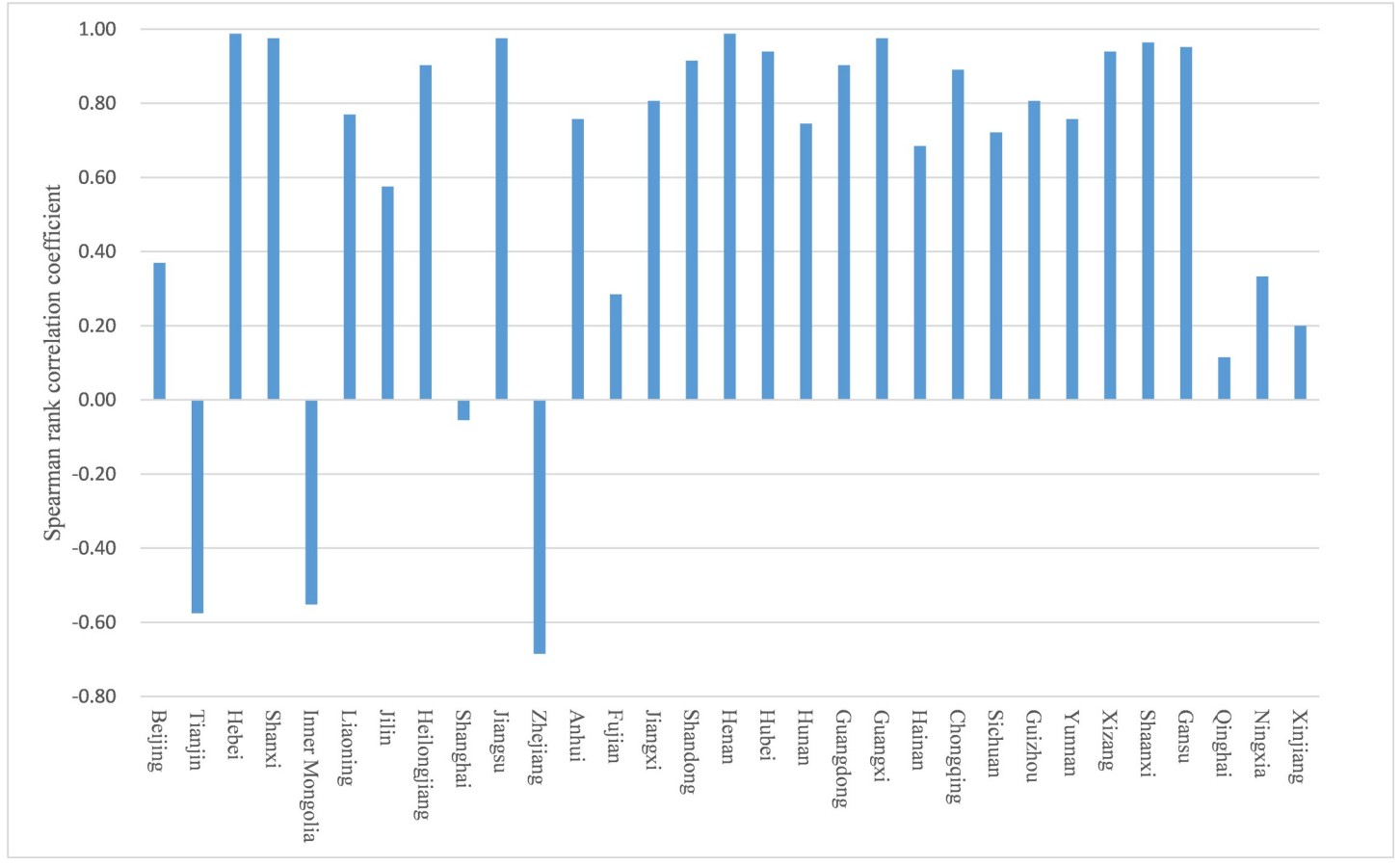

**Fig 4. Spearman rank correlation coefficient of the UTT level in each province.**

UTT level shows positive spatial autocorrelation, it is not significant after 2014, especially in 2019, which shows negative spatial autocorrelation characteristics. Under the economic spatial weighting matrix, the UTT level maintains a significant positive spatial autocorrelation at the 5% level. Therefore, Hypothesis H1 is valid under the economic spatial weighting matrix but not under the spatial weighting matrix after 2014. The possible reason is that the development of transportation and information technology overcomes the influence of geographical distance on knowledge exchange. Combining the advantages of the economic spatial weighting matrix mentioned above, the next step should choose a space panel model based on the economic spatial weighting matrix to analyze the influencing factors.

Fig 5 is Moran's I scatter plot of UTT levels in each province in 2019 under the economic spatial weighting matrix. The coordinate axis is divided into 4 quadrants centered on the mean. The first and third quadrants represent high-high (H-H) and low-low (L-L)

**Table 5. Moran's *I* of the UTT level in China.**

| Category | 2011 | 2012 | 2013 | 2014 | 2015 | 2016 | 2017 | 2018 | 2019 |
|---|---|---|---|---|---|---|---|---|---|
| Spatial weighting matrix | 0.054 *** | 0.044 ** | 0.043 ** | 0.008 | 0.004 | 0.010 | 0.020 | 0.004 | -0.014 |
| Economic spatial weighting matrix | 0.233 ** | 0.207 ** | 0.183 ** | 0.128 | 0.150 ** | 0.176 ** | 0.208 ** | 0.198 ** | 0.193 ** |

Note: *, **, *** indicate significance at the test levels of 10%, 5%, and 1%, respectively.

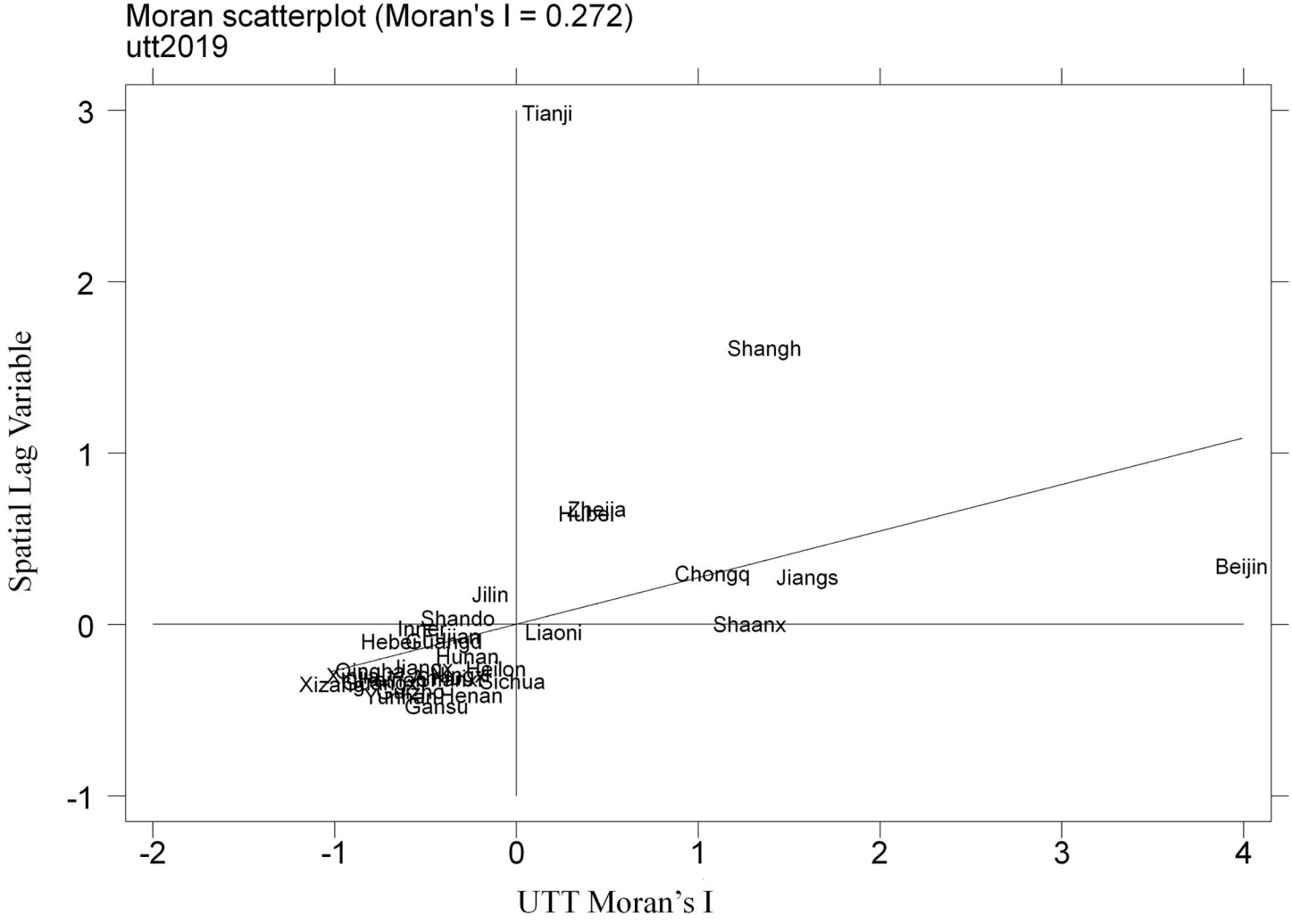

**Fig 5. Moran's I scatter plot of UTT levels in 2019.**

correlations, respectively, which indicates that the UTT levels of the province and its neighboring provinces are all very high or low. The provinces in the two quadrants have positive spatial autocorrelation. The second and fourth quadrants represent the spatial correlation method of low-high (L-H) and high-low (H-L), respectively, which indicates that the UTT level of this province and its neighboring provinces have an opposite trend, showing a negative spatial autocorrelation. It can be seen from Fig 3 that most provinces (28) are in the first or third quadrant, which further indicates that the UTT level of China shows a positive spatial autocorrelation and has a significant spatial spillover effect.

It can be further seen from Fig 5 that the UTTs of most provinces are at a low level and have a large space for improvement. Beijing, Jiangsu, Shanghai, Zhejiang, Tianjin, Hubei, Shaanxi, Chongqing and Liaoning have a high UTT level. These provinces are spread across four regions of China and have high levels of higher education. In addition, Shaanxi and Liaoning are in the fourth quadrant (H-L), indicating they have a higher UTT level, but the surrounding provinces have a lower UTT level, and the spatial spillover effect is not obvious.

## 5. Spatial econometric analysis of influencing factors

### 5.1 Spatial econometric model selection

Carrying out spatial econometric analysis of the UTT level, the premise is to select a suitable spatial econometric model. Both the explanatory variable and the explained variables have significant spatial autocorrelation. SDM is taken into consideration for analysis, which needs to be determined according to the test statistics.

The LM test is performed first, and the estimated result is shown in Table 6. The LM lag and the LM error statistic pass the 5% significance level test ($p<0.05$), which means the establishment of a spatial econometric model is more appropriate. The P values of the robust LM lag test and robust LM error test are both less than 1%, indicating that the LM test based on the nonspatial model accepts both SAR and SEM. Generally, SDM should be selected [64].

The estimated result of the Wald test is shown in Table 7, from which the Wald test statistics of SAR and SEM both pass the 1% significance level test. Therefore, SDM is more appropriate, which can decompose the direct effect and the indirect effect from the total effect and avoid the biased and inconsistent estimation results caused by the endogeneity of variables.

### 5.2 Spatial effect analysis of influencing factors

Based on the economic spatial weighting matrix, the fixed effect and random effect SDM are used to estimate the influencing factors of the UTT level, with the results shown in Table 8. The Hausman test result indicates that the fixed effect model should be used to guarantee the robustness of the analysis results.

The impact of each influencing factor on UTT can be concluded from Table 8. The results suggest that Hypothesis H3 is true, but H2 is partially true.

(1)  Government: Both the influence of government on the UTT of the province and neighboring provinces are negative. The UTT of the province decreases by 0.0335 units as government investment increases by 1 unit, which is not significant. The UTT of neighboring provinces decreases by 0.969 at a significance level of 1%. First, government input supports technological innovation and industrial development, which may seize the UTT market and have a crowding-out effect on UTT. Second, the government input on innovation promotes the development of the entire innovation ecosystem. Its investment and benefits are more reflected in enterprises and research institutes but have not been well reflected in the UTT field. Using the national R&D expenditures in 2019 as an example, the total R&D expenditures of enterprises and research institutes accounted for 90.3%, while universities accounted for only 8.1%. The proportion of expenditures invested in UTT is even smaller. In addition, there are some institutional barriers restricting UTT, so that government input cannot obtain good UTT output benefits.

(2)  Industry: The level of industrial development has a positive impact on the UTT level in the region and is significant at the 10% level. The UTT of the province increases by 0.0304 units as industrial development per 1 unit increases, indicating that a higher level of industrial

**Table 6. LM estimated results.**

| Test | T statistics | P value |
|---|---|---|
| LM lag | 3.9054 | 0.0481 |
| Robust LM lag | 7.7471 | 0.0054 |
| LM error | 4.4118 | 0.0357 |
| Robust LM error | 8.2535 | 0.0041 |

**Table 7. Wald test results.**

| Test | Statistics | P value |
|---|---|---|
| Wald test of spatial lag | 30.90 | 0.0001 |
| Wald test of spatial error | 28.20 | 0.0002 |

development has a stronger demand for technological innovation. The indirect effect is negative and significant at the 5% level, and the UTT of neighboring provinces decreases by 0.425 as industrial development increases by 1 unit, indicating that the level of industrial development harms the UTT of neighboring provinces. A possible reason is that industrial gatherings have a negative spatial spillover effect on the industrial development and techno-logical innovation of neighboring regions [65], thereby suppressing the UTT level.

(3) Financial Institute: The direct impact of financial development on UTT is negative and significant at the 5% level. The UTT of the province decreases by 0.0177 units in the wake of the financial development increases by 1 unit, while the UTT of neighboring provinces increases by 0.0471, which is not significant. This may be related to the fact that financial development in recent years has focused on virtual economies such as virtual real estate [66], while insufficient support for the real economy where UTT is located. In addition, the current financial system is not robust, and the financial scale is opposite to financial support. This seems to be a kind of "vanishing effect" [67]; financial deepening or private sector credit exceeds a certain size when the financial contribution to economic growth disappears. Financial development could harm economic growth since private credit exceeds 110% of GDP [68]. The role of financial development in economic growth has been found to have an inverted U-shaped relationship [69]. In other words, the expansion of the financial scale reduces the financial support effect [70–72].

(4) Technology Intermediary: The maturity of the technology market has positive effects on the UTT of provinces and negative effects on neighboring provinces but does not pass the significance test, indicating that technology intermediaries fail to have a significant positive

**Table 8. Estimated results of influencing factors.**

| Variable | Fixed Effect | Random Effect | Variable | Fixed Effect | Random Effect |
|---|---|---|---|---|---|
| Gov | -0.0335 (-1.09) | 0.0129 (0.29) | $W^* \cdot Gov$ | -0.9690*** (-2.65) | -0.5960* (-1.67) |
| Ins | 0.0304* (1.90) | 0.0302 (1.48) | $W^* \cdot Ins$ | -0.4250** (-2.09) | -0.3010 (-1.51) |
| Fin | -0.0177** (-2.55) | -0.0102 (-1.21) | $W^* \cdot Fin$ | 0.0471 (0.33) | 0.0256 (0.17) |
| Int | 0.0015 (0.04) | 0.0578 (1.26) | $W^* \cdot Int$ | -0.9920 (-1.00) | -0.4150 (-0.41) |
| Eco | 0.0137* (1.68) | 0.0083 (1.02) | $W^* \cdot Eco$ | 0.3360** (2.09) | 0.2220 (1.33) |
| Inf | 0.0498** (2.23) | 0.0569*** (2.83) | $W^* \cdot Inf$ | -0.4730* (-1.84) | -0.5140* (-1.81) |
| Open | -0.0391** (-2.13) | -0.0494*** (-2.59) | $W^* \cdot Open$ | 0.4200 (1.42) | 0.3120 (1.19) |
| $R^2$within | 0.4667 | 0.4359 | Hausman test | -15.88 | |
| Log-likelihood | 637.6443 | 533.5407 | | | |

Note: The Z statistics are in parentheses.

impact on provinces or a significant negative impact on neighboring provinces. Mature technology transfer channels, sound achievement evaluation mechanisms and professional technical manager teams can effectively promote UTT. Technology intermediaries are the link between universities and technology-demanding enterprises, creating value with their own professional services in the process of technology transfer. Technology intermediaries can facilitate the transformation of technology, promote the integration of industry-academia-research and advance the process of technology marketization, thus greatly improving the efficiency of UTT. However, the current technology market development is still in its infancy [73], and the improvement of technology intermediary capacity requires coordinated cooperation between government and other organizations. The impact of technology intermediaries on UTT is limited and has yet to be further strengthened.

(5) Economic: Both the influence of economic development on the UTT of the province and neighboring provinces are positive. The UTT of the province increases by 0.0137 units as economic development increases by 1 unit at a significance level of 10%, while the UTT of neighboring provinces decreases by 0.336 at a significance level of 5%, indicating that a higher level of regional economic development can promote the UTT level.

(6) Informatization: The regression coefficient of informatization is positive and significant at the 5% level, and the UTT of the province increases by 0.0498 units as informatization increases by 1 unit, indicating that informatization has a significant positive impact on UTT. While the impact on the UTT of neighboring provinces is negative, the UTT of the neighboring province decreases by 0.473 units in the wake of informatization increases by 1 unit at a significance level of 10%. Timeliness is very important to UTT. The high level of information can provide UTT with convenient information communication and reduce barriers caused by geographical distance.

(7) Openness: The degree of regional openness is significantly negative at the 5% level, and the UTT of the province decreases by 0.0391 units as openness increases by 1 unit, indicating that the technology introduction brought about by regional openness has a certain crowding-out effect on UTT. The impact on the UTT of neighboring provinces is positive, which is not significant.

## 5.3 Spatial effect heterogeneity analysis of influencing factors

According to Fig 3, the UTT level in the eastern region is significantly higher than that in other regions. Based on SDM under the economic spatial weighting matrix, the influencing factors of the UTT level in the eastern region and other regions are estimated and decomposed effects, respectively. The results are shown in Table 9. The direct effect reflects the influence of the change in the independent variable on the dependent variable in a certain area. The indirect effect, also known as the spatial spillover effect, is used to measure the impact of a certain explanatory variable in the "nearby" area.

Table 8 shows that the direct and indirect effects of the influencing factors in different regions are relatively heterogeneous. In terms of direct effects, government, technology intermediary, informatization, and openness are negative in the Eastern Region and positive in the Western Region. That is, when the UTT level is low, these factors have a positive effect on the UTT level; when the UTT level is high, these factors may simultaneously promote the independent innovation and technology introduction of enterprises and inhibit UTT. In addition, industry has a more obvious role in promoting UTT in other regions, and the level of regional economic development has a more obvious role in promoting UTT in the eastern region.

**Table 9. Estimation results of the spatial heterogeneity of various influencing factors.**

| Variable | Direct Effect | | Indirect Effect | |
|:---:|:---:|:---:|:---:|:---:|
| | Eastern Region | Other Regions | Eastern Region | Other Regions |
| Gov | -0.0683* | 0.0111 | 0.0157 | 0.0486 |
| | (-1.67) | (0.27) | (0.54) | (1.25) |
| Ins | 0.0079 | 0.0623*** | 0.0581* | 0.0312 |
| | (0.39) | (3.11) | (1.89) | (1.54) |
| Fin | -0.0424*** | -0.0119* | 0.0340 | 0.0031 |
| | (-2.85) | (-1.66) | (1.59) | (0.23) |
| Int | -0.1760*** | 0.2990** | -0.1070** | 0.0414 |
| | (-4.43) | (2.40) | (-2.46) | (0.34) |
| Eco | 0.0172** | 0.0083 | -0.0001 | -0.0139 |
| | (2.22) | (1.13) | (-0.00) | (-1.12) |
| Inf | -0.0881** | 0.0235 | 0.1200*** | -0.0460 |
| | (-2.18) | (1.29) | (3.63) | (-1.61) |
| Open | -0.0199 | 0.0860** | 0.0381*** | 0.0737 |
| | (-0.93) | (1.99) | (3.17) | (1.20) |

From the perspective of indirect effects, government and industry have spatial spillover effects in the eastern and other regions that are opposite to the national total spatial effect. The possible reason is that government input and industrial development level have a promotion effect on UTT in their respective regions, which can coordinate the development in the adjacent area, but it forms a siphon effect on the far area and inhibits UTT. For example, Jiangsu, Zhejiang and Shanghai in the eastern region are relatively closely connected, where industrial development provides a mutual market for UTT and has a significant promotion effect on UTT. However, it does not closely match the sci-tech achievements of universities in neighboring provinces such as Anhui and Jiangxi. In contrast, the market demand for UTT in these provinces is inhibited because of the output of industrial technology. The spatial effect of regional economic development is opposite to the previous two factors, showing significant spatial spillover across the nation but negative spatial spillover in the Eastern Region and other regions. Considering that the economic development level of the Eastern Region is higher than that of other regions, the larger regional economic development level among provinces can promote UTT, and vice versa. The technology intermediary has an obvious (significant at the 5% level) siphon effect in the Eastern Region and spatial spillover effects on other regions. The level of informatization is the opposite, with an obvious (significant at the 1% level) spatial spillover effect on the eastern region and negative spatial spillover in other regions.

# 6. Conclusion and enlightenment

## 6.1 Main conclusions

The paper establishes a comprehensive evaluation index system to measure provincial UTT levels from 2011 to 2019 in China. With the index system, the spatiotemporal distribution characteristics are analyzed, and the effects of influencing factors for the UTT level are estimated using SDM based on the economic spatial weighting matrix. The spatial heterogeneity effects of influencing factors from the perspective of effect decomposition are also analyzed. The main conclusions are as follows:

(1) The national UTT level is generally low, and the gap between provinces is prominent, showing a tiered distribution of high levels in the Eastern Region and low levels in the Western Region. Most provinces (27) show a growing trend of UTT levels.

(2) There is a significant spatial autocorrelation of the UTT level in the provinces. Most provinces have L-L correlations, while Beijing, Jiangsu, Shanghai, Zhejiang, Tianjin, Chongqing, and Hubei have H-H correlations.

(3) Industry, economy, and informatization have a significant role in promoting UTT, while financial institutes and openness have a significant inhibitory effect. Economy has a significant spatial spillover effect on UTT, while government, industry and informatization have a significant inhibitory effect on UTT in neighboring regions.

(4) The direct effects and indirect effects of influencing factors in the Eastern Region and other regions show significant spatial heterogeneity.

## 6.2 Policy enlightenment

Based on the research conclusions, we propose the following targeted suggestions:

(1) All provinces should formulate more realistic policies to promote UTT and improve the efficiency of sci-tech investment. Government input does not promote UTT, which further illustrates the "paradox" between low UTT levels and high input/output of technological innovation in universities. Considering the spatial heterogeneity effects of influencing factors, it is necessary to formulate detailed and targeted policies to improve the performance of R&D funding input. Using Beijing, where the UTT level is clearly at the leading position, as an example, it has formulated targeted policies to guide and support UTT work and overcome obstacles in the UTT process, including launching a special project for TTU platform construction, supporting the internal TTU institutions, carrying out evaluation and screening of sci-tech achievements and patent layout, strengthening the pilot test, maturation and docking of sci-tech achievements, and intensifying cooperation with socialized professional intermediaries. All the policies have achieved good results.

(2) UTT should be more oriented toward industrial development. Judging from the direct effects in the nation and other regions, the level of industrial development plays an important role in promoting UTT and has a certain role in the eastern region. From a practical point of view, disconnection from the industry is one of the main reasons for the low UTT level. Compared with enterprises, the technological achievements of universities lack competitiveness in the technology market, and it is difficult for them to meet the needs of industrial development. University innovation should be guided by the needs of economic and social development and should focus on facing industrial needs from the beginning of basic research to solve the key common technology problems that restrict industrial development. Universities can adopt measures to strengthen the connection and docking with industries and improve the industrial fit of the innovation in universities. The measures include establishing industrial research institutes, building an industry-university-research alliance, improving the training requirements of research personnel, inviting enterprises to participate in the application of research projects, and cultivating a professional talent team for TTU.

(3) All provinces should create a good innovation ecosystem. Each actor in the ecosystem has a certain impact on UTT. The government should increase investment and formulate more targeted policies in technology transfer, guide financial institutes to "depart from the virtual to the real" and serve the real economy and high-tech industries, increase the construction of technology intermediaries and markets and create a good innovation ecological environment.

## Supporting information

**S1 Data.**
(XLSX)

**S1 File.**
(PDF)

## Author Contributions

**Conceptualization:** Haining Fang, Qing Yang.

**Data curation:** Haining Fang, Xingxing Liu.

**Formal analysis:** Haining Fang.

**Funding acquisition:** Haining Fang, Qing Yang.

**Investigation:** Jinmei Wang, Lanjuan Cao.

**Methodology:** Haining Fang, Qing Yang.

**Project administration:** Haining Fang.

**Resources:** Qing Yang.

**Software:** Haining Fang, Qing Yang, Xingxing Liu.

**Supervision:** Haining Fang, Qing Yang.

**Validation:** Jinmei Wang.

**Visualization:** Qing Yang, Xingxing Liu.

**Writing – original draft:** Haining Fang.

**Writing – review & editing:** Jinmei Wang, Qing Yang, Lanjuan Cao.

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
