## [Decision Letter · Decision Letter 0]

22 Dec 2021

PONE-D-21-32734Spatiotemporal characteristics and influence factors analysis of universities technology transfer level in China: the perspective of innovation ecosystemPLOS ONE

Dear Dr. Wang,

Thank you for submitting your manuscript to PLOS ONE. After careful consideration, we feel that it has merit but does not fully meet PLOS ONE’s publication criteria as it currently stands. Therefore, we invite you to submit a revised version of the manuscript that addresses the points raised during the review process.

ACADEMIC EDITOR

Dear author,

Thank you for submitting your paper to PLOS ONE.

I am delighted to invite you to revise and resubmit your paper. However, please note that reviewers are very critical towards the methodological approach of this study, and pose several major shortcomings and points of improvement that you need to pay close attention to. Reviewer’s detailed comments (listed below) are self-explanatory, so there is no need to reiterate them here.

Please note that you should provide a point-to-point response where you discuss how you changed the paper based on the comments, or alternatively a justified rebuttal why you could not follow a particular advice.

I am looking forward to receiving your revised manuscript.

Sincerely,

Stefano Ghinoi

We look forward to receiving your revised manuscript.

Kind regards,

Stefano Ghinoi, Ph.D.

Academic Editor

PLOS ONE

Journal Requirements:

 “This research was funded by the 2020 Social Science Foundation Key Project of Hubei Province (New Think Tank Project) (No. BSKZD2020002) and 2020 Universities Practical Education Project of Hubei Province (No. 2020SJJPE3004).”

Reviewers' comments:

Reviewer's Responses to Questions

**Comments to the Author**

1. Is the manuscript technically sound, and do the data support the conclusions?

Reviewer #1: No

Reviewer #2: Partly

2. Has the statistical analysis been performed appropriately and rigorously? 

Reviewer #1: No

Reviewer #2: Yes

3. Have the authors made all data underlying the findings in their manuscript fully available?

Reviewer #1: Yes

Reviewer #2: No

4. Is the manuscript presented in an intelligible fashion and written in standard English?

Reviewer #1: No

Reviewer #2: No

5. Review Comments to the Author

Reviewer #1: Dear colleagues,

Please let me congratulate for your work. It is wise to look from different angle to TTOs. I am very impress about the idea and approach. However, I do have some concerns, which are major and put me in hesitation between major revision and rejection. However, I am sure you can clarify issues, which put me in hesitation.

1- Please be clear about the research gap you are addressing. What is the aim of this paper and what is the missing in the literature and what is you are trying to clarify. Unfortunately, this is not easy to find out from text. Why don’t you be clear about the research gap and your research question?

2- Your literature review needs to be more focus in line with your research question. For instance, innovation ecosystem is not main topic of the paper but you have a section for it? Why, can’t you just summarize it in a paragraph.

3- Please develop hypothesises (if you can) and link them to literature and your research model. Pls also draw a research model that can be useful to flow of your research logic.

4- Your methodology section is not very easy to follow and to understand. First, you used different models but no explanation why you have used them. Please elaborate on that. Also, create a smooth link between each section of methodology section.

5- Please also explain why China has been chosen for this research. What makes China interesting for this study.

6- Please try to update literature more with recent work. You can benefit from following new publications;

https://www.sciencedirect.com/science/article/abs/pii/S0160791X21001408

https://ieeexplore.ieee.org/abstract/document/9418622

https://www.sciencedirect.com/science/article/pii/S0160791X21002578

Reviewer #2: Referee report Manuscript PONE- D-21-32734

The topic is interesting, and the paper's contribution to the recent literature is clear. I feel that the manuscript would benefit from additional work to improve the statistical analysis and its presentation.

In more detail:

i) The paper would benefit from review by a native English speaker to assist with sentence construction and spelling. Examples of sentences not correct/clear:

- Contrary to expectation, the rate of commercialization 42 of university inventions to be too

Low (line 42)

- “but it cannot fully reflect mutual influence of various regions based economic” (p.12. line 8)

ii) Among the regressors, the author(s) should include one (or more) variable(s) observing the specialization of the provinces’ economies

iii) The author(s) should provide more technical details about the evaluation index screening and weighting used to build the UTT measure.

iv) Is the variable “Actual income in the year of UTT/Thousand yuan” (tab. 1) measured at constant prices? It should be. Otherwise, the UTT index might display a growing trend because of inflation.

v) Please provide references for using the spearman rank correlation coefficient as a tool to examine UTT trends

vi) please generate figures’ captions and make figures more clear by introducing clear labels for the x and y axes.

vii) Please clarify that the analysis only allows the observation of robust ceteris paribus correlations among the variables included in the analysis.

viii) The author(s) focus their comments on statistical significance results. Instead, it would be essential to provide detailed comments about the magnitude of the coefficients calculated through the regression analysis. Commenting on the coefficients will provide essential elements about the economic significance of the findings.

ix) The interpretation of the results concerning the financial development variable requires more argument (p.22).

x) The interpretation of the results concerning “Technology intermediary” is unclear to me. To the best of my understanding, the variable is not statistically significant, while the comment assumes it is significant.

xi) “The possible reason is that the influence of these factors on UTT has an inverted U-shaped threshold effect”. (p. 24, line 4). Please provide a more precise illustration of this interpretation. To the best of my understanding, the findings demonstrate that some of the factors' effects vary according to the dependent variable's level.

Minor comments

- please provide details about how the effective patent application rate (line 49 is calculated

- The analysis allows inspecting the correlates of UTT in the Chinese provinces while it cannot provide any helpful information about “the driving mechanism of influencing factors” (line 58)

- the “reference source not found” errors appears at p. 4

- “most of them” (p. 6 line 9); most of who?

6. PLOS authors have the option to publish the peer review history of their article (what does this mean?). If published, this will include your full peer review and any attached files.

Reviewer #1: **Yes: **Serdal Temel

Reviewer #2: No

---

## [Author Response · Author response to Decision Letter 0]

15 Feb 2022

Response to Reviewer Comments

We would like to express my appreciation to you for suggesting how to improve our paper. We hope that the revised manuscript is now suitable for publication.

For reviewers’ comments, we have done following corresponding explanations and revisions.

Reviewer 1

Point 1. Please be clear about the research gap you are addressing. What is the aim of this paper and what is the missing in the literature and what is you are trying to clarify. Unfortunately, this is not easy to find out from text. Why don’t you be clear about the research gap and your research question?

Response 1: Thank you for your advice. We are very sorry that the research purpose of this paper is not clear in section 1. We have reorganized the content of introduction, enriched the literature review on UTT influencing factors, introduced the research gap of current research, and further clarified the research question and aim. (p. 4-5, line 66-108)

Point 2. Your literature review needs to be more focus in line with your research question. For instance, innovation ecosystem is not main topic of the paper but you have a section for it? Why, can’t you just summarize it in a paragraph.

Response 2: Thank you for your advice. We are sorry that the literature review is not in line with the research question. Under your advice, we have readjusted the structure of the article. On the one hand, we have reorganized the structure of introduction, moved the literature review to the section and added references on UTT level influencing factors, to make the literature review more focus in line with the research question. (p. 4-5, line 66-108) On the other hand, we have compressed the literature review of innovation ecosystem into a paragraph and moved it to the section of method together with the influencing factors of UTT (the initial manuscript 2.3). (p. 11-13, line 213-272)

Point 3. Please develop hypothesises (if you can) and link them to literature and your research model. Pls also draw a research model that can be useful to flow of your research logic.

Response 3: Thank you for your advice. We have added methodology model shown as Figure 1 to flow of the research logic. (p. 6, line 114-120) We are very sorry for it is difficult to put forward appropriate hypotheses because of the role of influencing factors different in neighboring provinces and few relevant research conclusions.

Point 4. Your methodology section is not very easy to follow and to understand. First, you used different models but no explanation why you have used them. Please elaborate on that. Also, create a smooth link between each section of methodology section.

Response 4: Thank you for your advice. We have added methodology model shown as Figure 1 and a paragraph to introduce the method to flow of the research logic. (p. 6, line 114-120) Before the introduction of UTT evaluation index method, a summary of evaluation methods has been added to illustrate the necessity of selecting comprehensive evaluation method. The steps of index screening and weighting method based on information contribution rate have been added to make the methodology part easier to understand. (p. 6-8, line 122-152) In addition, we have moved the influencing factors of UTT (the Initial manuscript 2.3) before the spatial econometric model of influencing factors, making them integrated and easy to understand. (p. 11-13, line 213-272)

Point 5. Please also explain why China has been chosen for this research. What makes China interesting for this study.

Response 5: Thank you for your advice. In section 1, we have added the reasons for choosing China for research. Over the past three decades China has made unprecedentedly large investments in university research and Chinese universities are now becoming major contributors to the global scientific community. However, it can be seen from the data that the proportion of UTT is still too low not to match its level of economic development. As the second largest university research system in the world, a better understanding of the complex ways of Chinese UTT is critical for scholars and policy makers globally. In the same way, improving Chinese UTT level can enhance innovation vitality and further promote economic development. Therefore, studies on UTT influencing factors are of great significance. (p. 3-4, line 57-65) China has a vast territory, and the economic development levels of various provinces are quite different. Due to the spatial heterogeneity of the impact of various factors on UTT, the current analysis of the influencing factors of UTT in China is insufficient. There is no relevant literature on spatiotemporal analysis of UTT level in China. (p. 5, line 101-14)

Point 6. Please try to update literature more with recent work. You can benefit from following new publications;

https://www.sciencedirect.com/science/article/abs/pii/S0160791X21001408

https://ieeexplore.ieee.org/abstract/document/9418622

https://www.sciencedirect.com/science/article/pii/S0160791X21002578

Response 6: Thank you for your advice. We have carefully studied the literature you listed, benefited a lot from it and quoted it in this article. These literatures also have important guiding significance for future research. (references [19,20,26])

 

Reviewer 2

Point 1. The paper would benefit from review by a native English speaker to assist with sentence construction and spelling. Examples of sentences not correct/clear:

- Contrary to expectation, the rate of commercialization 42 of university inventions to be too Low (line 42)

- “but it cannot fully reflect mutual influence of various regions based economic” (p.12. line 8)

Response 1: Thank you for your advice. We are very sorry for the errors in sentence and spelling making for not pleasant reading. We have revised the two problems you pointed out and asked a native English speaker to proofread.

The sentence “Contrary to expectation, the rate of commercialization 42 of university inventions to be too Low (line 42)” has been modified to “Over the past 20 years, universities have made steady progress in their efforts to facilitate the technology transfer process by working with industry [7], but UTT level is still remarkably low.” (p.3, line 44-45)

The sentence “but it cannot fully reflect mutual influence of various regions based economic” (p.12. line 8) has been modified to “but it does not sufficiently reflect the interaction of economic bases across regions”. (p.10, line 194-195)

Point 2. Among the regressors, the author(s) should include one (or more) variable(s) observing the specialization of the provinces’ economies

Response 2: Thank you for your advice. We are very sorry for not clarifying the point. Among the regressors, we used 3 indexes including GDP, high-tech industry development and financial development to observe the specialization of the provinces’ economies. (p. 12-13,line 248-271)

Point 3. author(s) should provide more technical details about the evaluation index screening and weighting used to build the UTT measure.

Response 3: Thank you for your advice. We are very sorry for not clarifying the technical details. We have added the steps of index screening and weighting method based on information contribution rate. Considering the repeatability factor of the paper, detailed calculation process can be found in the related reference [41]. (p. 7-8, line 140-150)

Point 4. Is the variable “Actual income in the year of UTT/Thousand yuan” (tab. 1) measured at constant prices? It should be. Otherwise, the UTT index might display a growing trend because of inflation.

Response 4: Thank you for your advice. We are very sorry for not clarifying the point. “Actual income in the year of UTT/Thousand yuan” measured at constant prices and the original data has been processed, which has been explained in the paper. (p. 8, line 150-152)

Point 5. Please provide references for using the spearman rank correlation coefficient as a tool to examine UTT trends

Response 5: Thank you for your advice. We are very sorry for the omission of references and have added references for using the spearman rank correlation coefficient as a tool to examine UTT trends. (p. 8, line 161)

Point 6. please generate figures’ captions and make figures more clear by introducing clear labels for the x and y axes.

Response 6: Thank you for your advice. We are very sorry for the missing graphics label. We redrew figures 3 and 4, and added ordinate labels to make the figures clearer. 

Point 7. Please clarify that the analysis only allows the observation of robust ceteris paribus correlations among the variables included in the analysis.

Response 7: Thank you for your advice. We are very sorry for not clarifying the point. The problem of achievement transformation involved in this paper is a kind of exploratory problem, not a traditional economic forecasting problem. The fixed effect model is recommended for the estimation results, and its regression can obtain consistent results with strong robustness. Due to space limitation, we have modified the expression of robustness in this paper and will continue to study the transformation conditions to verify robustness in the future. (p. 20, line 375-376)

Point 8. The author(s) focus their comments on statistical significance results. Instead, it would be essential to provide detailed comments about the magnitude of the coefficients calculated through the regression analysis. Commenting on the coefficients will provide essential elements about the economic significance of the findings.

Response 8: Thank you for your advice. We are very sorry for not clarifying the point. Since the paper did not adopt normalization treatment for various index data, we did not compare the regression coefficients of various influencing factors. We will try to conduct comparative analysis on the regression coefficients of each variable in the future research.

Point 9. The interpretation of the results concerning the financial development variable requires more argument (p.22).

Response 9: Thank you for your advice. We are very sorry for not clarifying the point. This paper adds the literature on explaining the results of financial development variables. The direct impact of financial development on UTT is negative and significant at the level of 5%, which may be related to the fact that financial development in recent years has focused on virtual economies such as virtual real estate, insufficient support for the real economy where UTT is located. In addition, the current financial system is not robust and the financial scale is opposite to financial support, the expansion of financial scale reduces the financial support effect [66-68]. (p. 22-23, line 398-403)

Point 10. The interpretation of the results concerning “Technology intermediary” is unclear to me. To the best of my understanding, the variable is not statistically significant, while the comment assumes it is significant.

Response 10: Thank you for your advice. We are very sorry for not clarifying the point. This paper explains the difference between the result and hypothesis of " Technology Intermediary ": because the current technology market development is still in its infancy [69], the support to UTT is still limited. (p.23, line 404-408)

Point 11. “The possible reason is that the influence of these factors on UTT has an inverted U-shaped threshold effect”. (p. 24, line 4). Please provide a more precise illustration of this interpretation. To the best of my understanding, the findings demonstrate that some of the factors' effects vary according to the dependent variable's level.

Response 11: Thank you for your advice. We are very sorry for not clarifying the point. The inverted U-shaped conclusion in the paper is the author's estimate, without rigorous data demonstration. In consideration of the rigor of the article, we deleted the description of inverted U shape and carried out further research after consideration. 

Minor comments

- please provide details about how the effective patent application rate (line 49 is calculated

Response 12: Thank you for your comments. We are very sorry for the wrong word. The effective patent application rate in line 49 should be corrected to the effective patent application rate. The data comes from “China Intellectual Property Survey Report 2019” by China National Intellectual Property Administration. (p. 3, line 52-53)

- The analysis allows inspecting the correlates of UTT in the Chinese provinces while it cannot provide any helpful information about “the driving mechanism of influencing factors” (line 58)

Response 13: Thank you for your comments. We are very sorry for the inaccurate expression. We changed “the driving mechanism of influencing factors” to “the impact of various factors on UTT”. (p.6, line 108)

- the “reference source not found” errors appears at p. 4

Response 14: Thank you for your comments. We are very sorry for the error at p.4. We rearranged and checked the reference labels of the references.

- “most of them” (p. 6 line 9); most of who?

Response 15: Thank you for your comments. We are very sorry for not clarifying the point. We have asked a native English speaker to help us to assist with sentence construction and spelling.

The phrase “and most of them” has been modified to “most of which”, referring to the evaluation of UTT capability. (p.6, line 122-123)

---

## [Editor Report · Decision Letter 1]

9 Mar 2022

PONE-D-21-32734R1Spatiotemporal characteristics and influencing factors analysis of universities technology transfer level in China: the perspective of innovation ecosystemPLOS ONE

Dear Dr. Wang,

Thank you for submitting your manuscript to PLOS ONE. After careful consideration, we feel that it has merit but does not fully meet PLOS ONE’s publication criteria as it currently stands. Therefore, we invite you to submit a revised version of the manuscript that addresses the points raised during the review process. Please find my comments in the "Additional Editor Comments" section below.

We look forward to receiving your revised manuscript.

Kind regards,

Stefano Ghinoi, Ph.D.

Academic Editor

PLOS ONE

Additional Editor Comments:

Dear Authors,

Thanks for revising your paper following the indications suggested by the reviewers. Your efforts in trying to improve your work are evident, but there are still some major issues that need to be addressed.

I am still struggling to understand what is the research gap in the literature. According to what you wrote, “there is no relevant literature on spatiotemporal analysis of UTT level in China”. However, this is based on a review of the literature that is missing some relevant recent studies:

• Yuan et al. (2018). Dynamic capabilities, subnational environment, and university technology transfer

• Wu et al. (2021). Does social trust stimulate university technology transfer? Evidence from China

• Li and Tang (2021). A dynamic capabilities perspective on pro-market reforms and university technology transfer in a transition economy

Point 2 from Reviewer#1 did not suggest deleting entirely your literature review section, but only to reduce section 2.1. You should keep section 2 as your literature review section, and strengthening it with more studies on UTT in China – in order to finetune and narrow your work, and create a better linkage between research gap, research question, literature, and methodological approach

As a result of the removal of the whole literature review section, actually section 2.3 includes a mix of method and literature review. This is rather confusing.

You have not addressed one key issue raised by Reviewer#1:

• Point 3: your reply “it is difficult to put forward appropriate hypotheses because of the role of influencing factors different in neighboring provinces and few relevant research conclusions” is not satisfactory: hypotheses are based on the literature, not on the presence different influencing factors in different provinces. Moreover, why this should be a problem in the development of hypotheses?

You did not address some of the key issues raised by Reviewer#2:

• Point 2: The specialization of the provinces’ economies should be based on a survey on the local enterprises based on their NACE sectors

• Your reply to Point 8 is not satisfactory: you do not have to compare the coefficients of the variables, you need to comment on the meaning (in economic and statistical terms) of the single coefficients

• Point 9: The interpretation of the results concerning the financial development variable is still not sufficient

• Your response to Point 10 is unclear and does not address the comment from the reviewer

Minor points:

• Line 36: the acronym for UTT is described in the abstract, but not in the main text. The first time you use it, you need to clarify what the acronym stands for

• Line 37 and 46: I do not understand why citing twice the paper of Trippl et al. (2015) in parenthesis

• The first section in the Introduction is based on several citations from the paper of Trippl et al. (2015): this is rather unusual

• Line 254: what does it mean that the UTT evaluation method is based on “the basis of the full-text analysis”?

• When presenting Table 2, you should include a column with a short description of the variables and their unit of measure
---

## [Author Response · Author response to Decision Letter 1]

13 Apr 2022

Response to Comments

We would like to express my appreciation to you for suggesting how to improve our paper. We hope that the revised manuscript is now suitable for publication.

For additional editor and reviewers’ comments, we have done following corresponding explanations and revisions.

Additional Editor Comments:

Comment 1: I am still struggling to understand what is the research gap in the literature. According to what you wrote, “there is no relevant literature on spatiotemporal analysis of UTT level in China”. However, this is based on a review of the literature that is missing some relevant recent studies:

• Yuan et al. (2018). Dynamic capabilities, subnational environment, and university technology transfer

• Wu et al. (2021). Does social trust stimulate university technology transfer? Evidence from China

• Li and Tang (2021). A dynamic capabilities perspective on pro-market reforms and university technology transfer in a transition economy

Response 1: Thank you for pointing out the shortcomings. We have carefully studied the literature you recommended and conducted some literature searches to put forward the research gaps and innovations.

The following discussion have been added in the revised manuscript here for your easy reference: Some scholars have noted the heterogeneous impact of subnational environment [15], social trust [16], and pro-market reforms [17] on UTT. Hou et al. discover that the moderating role of intermediary organizations in academia-industry cooperation and industrial innovation varies in inland and coastal areas [18]. The sustainable development and transformation of knowledge and innovation should be considered as a dynamic and systematic coevolutionary process [19]. Considering spatial relationship in regression analysis becomes a popular paradigm. The heterogeneous impact of regional innovation ecosystems on UTT need to be recognized deeper with spatial analysis as well. The innovations and contributions of this paper are: (1) establishing a comprehensive evaluation index system of UTT level based on the evaluation index screening and weighting method based on information contribution rate; (2) analyzing the heterogeneous impact of innovation actors on UTT from the perspective of innovation ecosystem. (p.4, line 67-77)

Comment 2: Point 2 from Reviewer#1 did not suggest deleting entirely your literature review section, but only to reduce section 2.1. You should keep section 2 as your literature review section, and strengthening it with more studies on UTT in China – in order to finetune and narrow your work, and create a better linkage between research gap, research question, literature, and methodological approach

As a result of the removal of the whole literature review section, actually section 2.3 includes a mix of method and literature review. This is rather confusing.

Response 2: Thank you for your guidance. In the last revision, we referred to some literature structures and decomposed the literature review part into the introduction and methods parts. As you point out, we have recognized that such a structure is not common, and restored the literature review section. At the same time, we have added some literatures in the literature review and proposed hypotheses based on the literature review. (p.4-8, line 82-167)

Comment 3: You have not addressed one key issue raised by Reviewer#1:

• Point 3: your reply “it is difficult to put forward appropriate hypotheses because of the role of influencing factors different in neighboring provinces and few relevant research conclusions” is not satisfactory: hypotheses are based on the literature, not on the presence different influencing factors in different provinces. Moreover, why this should be a problem in the development of hypotheses?

Response Point 3: Thanks for your guidance. We have seriously reflected on your suggestions, put forward hypotheses in the literature review section, and explained whether the hypotheses are valid in the result analysis.

The following hypotheses have been added in the revised manuscript here for your easy reference:

H1: Provincial UTT Level shows spatial autocorrelation in China; H2: The factors in innovation ecosystem have positive impact on UTT; H3: The influence of the factors on UTT level is spatially heterogeneous in China. (p.6, line 113, p.8, line 116-117, p17, line301-304, and p21, line350-351)

Comment 4: You did not address some of the key issues raised by Reviewer#2:

• Point 2: The specialization of the provinces’ economies should be based on a survey on the local enterprises based on their NACE sectors

Response Point 2: Thank you for your advice. According to your suggestion, we have searched literatures related to university technology transfer and economic specialization, only a small amount of literature addressed both keywords, such as University technology transfer and the evolution of regional specialization_ the case of Turin，Co-evolution patterns of university patenting and technological specialization in European regions. These studies build a framework of the co-evolution of university technology transfer and regional specialization, and we are deeply inspired and provide good ideas for further in-depth research of the interaction between innovation ecosystem and university technology transfer. Moreover, these studies also prove the necessity of heterogeneity analysis of the influencing factors of university technology transfer. For example, University technology transfer and the evolution of regional specialization the case of Turin states: The geographical closeness between universities and firms is important because the exchanged knowledge is cumulative, localized, and tacit in nature allowing local firms to access the results of academic research more easily. Geographical proximity may also strengthen other forms of proximity such as cognitive, organizational, and technological closeness. Other literatures have also reached some similar conclusions, such as the absorptive capacity of firms may affect the effective transfer of knowledge from universities to local firms, the transfer of complex knowledge is affected by the technological proximity of economic agents within the local system. However, after careful discussion and research, we believe that economic specialization (or regional specialization) is not the main body of the innovation ecosystem in the framework of this paper, nor is it an essential influencing factor. Therefore, the paper does not regard economic specialization as research variable for the time being.

In addition, we refer to the relevant variable classifications of classified statistical agencies such as the National Bureau of Statistics of China and the State Administration for Market Regulation (refer to the website: http://www.stats.gov.cn/tjsj/ndsj/2019/indexeh.htm)

• Your reply to Point 8 is not satisfactory: you do not have to compare the coefficients of the variables, you need to comment on the meaning (in economic and statistical terms) of the single coefficients

Response Point 8: Thank you for your advice. We are very sorry for not clarifying the coefficients meaning. According to your suggestion, we have supplemented the analysis of the meaning of the single coefficients of the estimated influencing factors in Table 8. 

The following discussion have been added in the revised manuscript here for your easy reference: The UTT of the province decreases by 0.0335 units along with the government investment increases by 1 unit, which is not significant. While, the UTT of neighboring provinces decreases by 0.969 at a significant level of 1%; The direct impact of financial development on UTT is negative and significant at the level of 5%. The UTT of the province decreases by 0.0177 units in the wake of the financial development increases by 1 unit, while the UTT of neighboring provinces increases by 0.0471, which is not significant. (p.21-24, line 353-355, line 365-366,line 368, line 374-376, line 399-401, line 404-407)

• Point 9: The interpretation of the results concerning the financial development variable is still not sufficient

Response Point 9: Thank you for your advice. We are very sorry for not clarifying the interpretation of the results concerning the financial development variable. To better explain the mechanism of action of financial development, we have added three literatures illustrating the opposite of financial size and financial support. 

The following explanations have been added in the revised manuscript here for your easy reference: This seems to be kind of “Vanishing Effect” [68], financial deepening or private sector credit exceeds a certain size when the financial contribution to economic growth disappears. Financial development could have a negative impact on economic growth since private credit exceeds 110% of GDP[69]. The role of financial development in economic growth has been found as an inverted U-shaped relationship[70]. (p.22, [68-70],line 379-383)

• Your response to Point 10 is unclear and does not address the comment from the reviewer

Response Point 10: Thank you for your advice. We are very sorry for not clarifying the point. In table 8, Int variable (Technology Intermediary) has positive effects on UTT of province and negative effects on neighboring provinces, but does not pass the significance test, indicating that technology intermediary fails to have a significant positive impact on province nor significant negative impact on neighboring provinces. On the one hand, the value creation capability of technology intermediaries is clearly. Mature technology transfer channels, sound achievement evaluation mechanism and professional technical manager team can effectively promote UTT. Technology intermediaries are the link between universities and technology-demanding enterprises, creating value with their own professional services in the process of technology transfer. Technology intermediaries can facilitate the transformation of technology, promote the integration of industry-academia-research and advance the process of technology marketization, thus greatly improving the efficiency of UTT. On the other hand, various factors constrain the role played by technology intermediaries in the UTT process. The current technology market development is still in its infancy [74] and the improvement of technology intermediary capacity requires coordinated cooperation between government and other organization, the support to UTT is still limited, the impact of technology intermediary on the UTT has yet to be further strengthened. (p.22-23, line 385-397)

 

Minor points:

• Line 36: the acronym for UTT is described in the abstract, but not in the main text. The first time you use it, you need to clarify what the acronym stands for

Response 1: Thank you for your advice. We are very sorry for this. In introduction where we first use it, we have added specific meaning of the acronyms. (p.2, line 32)

• Line 37 and 46: I do not understand why citing twice the paper of Trippl et al. (2015) in parenthesis

• The first section in the Introduction is based on several citations from the paper of Trippl et al. (2015): this is rather unusual

Response 2: Thank you for your advice. We are very sorry for not clarifying the point. In the revise revision “Revised Manuscript with Track Changes”, due to some technical factors (using word's revision mode), there were errors in the marking of references, most of them were marked as [1], but these are different literatures, and the new revised version will not have this problem again.

• Line 254: what does it mean that the UTT evaluation method is based on “the basis of the full-text analysis”?

Response 3: Thank you for your advice. We are very sorry for not clarifying the point. This expression is kind of chronic statement with no specific meaning, so we have deleted this sentence. Sorry for confusing you.

• When presenting Table 2, you should include a column with a short description of the variables and their unit of measure

Response 4: Thank you for your advice. We are very sorry for not clarifying the point. According to your suggestion, we have added Table 2. (p.14, line 273)

The following is Table 2 have been added in the revised manuscript here for your easy reference:

Table 2. Proxy Variables for Influencing Factors

Influence Factor Proxy Variable

UTT Level (UTT) Results by the evaluation index screening and weighting method based on information contribution rate

Government (Gov) R&D investment intensity

Industry (Ind) The operating income of high-tech industry per 10,000 people (10, 000 yuan)

Financial Institute (Fin) The ratio of balance of deposits to loans of financial institutions at the end of the year to GDP

Technology Intermediary (Int) technology market turnover per capita (10, 000 yuan)

Economic (Eco) per capita GDP (10, 000 yuan)

Informatization (Inf) the number of Internet interfaces per 10,000 people

Openness (Open) the investment amount of foreign-invested enterprises per 10,000 people (10, 000 dollars)

Closing comment to Editor: We hope the revised manuscript is now acceptable to you. If not, we are glad to receive any further feedback which we shall continue to apply our best effort to address.

---

## [Editor Report · Decision Letter 2]

26 Apr 2022

PONE-D-21-32734R2Spatiotemporal characteristics and influencing factors analysis of universities technology transfer level in China: the perspective of innovation ecosystemPLOS ONE

Dear Dr. Wang,

Thank you for submitting your manuscript to PLOS ONE. After careful consideration, we feel that it has merit but does not fully meet PLOS ONE’s publication criteria as it currently stands. Therefore, we invite you to submit a revised version of the manuscript that addresses the points raised during the review process.

 In particular, your manuscript needs a proofreading from a professional translator. Some sentences are really hard to understand and there are several typos in the text.

We look forward to receiving your revised manuscript.

Kind regards,

Stefano Ghinoi, Ph.D.

Academic Editor

PLOS ONE
---

## [Author Response · Author response to Decision Letter 2]

9 May 2022

Response to Comments

We would like to express my appreciation to you for suggesting how to improve our paper. We hope that the revised manuscript is now suitable for publication.

For Academic Editor’ comments, thank you for your valuable and thoughtful comments. We have carefully checked and improved the English writing in the revised manuscript. For Journal Requirements, we have done following corresponding explanations and revisions.

Journal Requirements:

Thank you for your advice. According to your suggestion, we have checked reference list and refined the content and format according to the requirements of the journal. We have deleted Reference 54, because of repeat with Reference 52. The specific modifications are as follows.

Reference 1: Added magazine abbreviations and article Issue.

Trippl M, Sinozic T, Smith HL. The role of universities in regional development: Conceptual models and policy institutions in the UK, Sweden and Austria. Eur Plan Stud. 2015; 23(9): 1722-1740.

Reference 3: Added article specific time of publication.

Han J. Technology Commercialization through Sustainable Knowledge Sharing from University-Industry Collaborations, with a Focus on Patent Propensity. Sustainability. 2017 Oct 8. doi: 10.3390/su9101808.

Reference 4: Added magazine abbreviations.

Etzkowitz H, Leydesdorff L. The dynamics of innovation: From national systems and “mode 2” to a triple helix of university–industry–government relations. Res Policy. 2000; 29(2):109-123.

Reference 5: Added magazine abbreviations.

Harman G． Australian university research commercialization: perceptions of technology transfer specialists and science and technology academics．J High Educ Policy Manag. 2010;32(1):69-83．

Reference 6: Added magazine abbreviations and article Issue.

Buenstorf G． Is commercialization good or bad for science? Individual - level evidence from the Max Planck Society．Res Policy. 2009; 38(2):281-292．

Reference 7: Changed article published year.

Muscio A. What drives the university use of technology transfer offices? Evidence from Italy. J Technol Transf. 2009; 35(2): 181-202.

Reference 9: Added magazine abbreviations.

Markman GD, Phan PH, Balkin DB, Gianiodis PT. Entrepreneurship and university-based technology transfer. J Bus Ventur. 2005;20(2):241-263.

Reference 10: Added magazine abbreviations.

Sampat B N. Patenting and US academic research in the 20thcentury: The world before and after Bayh-Dole. Res Policy. 2006;35(6):772-789．

Reference 13: Changed citation format and added article Issue.

Chen, A., Patton, D. & Kenney, M. University technology transfer in China: a literature review and taxonomy. J Technol Transf. 2016; 41(5): 891-929.

Reference 14: Added authors and magazine abbreviations.

Yang W, Yu X, Wang D, Yang JR, Zhang B. Spatio-temporal evolution of technology flows in China: patent licensing networks 2000-2017. J Technol Transf. 2021,46(5):1674-1703.

Reference 15: Added authors and magazine abbreviations.

Yuan CH, Li Y, Vlas CO, Peng MW. Dynamic capabilities, subnational environment, and university technology transfer. Strateg Organ. 2018; 16(1):35-60.

Reference 16: Added magazine abbreviations and article doi.

Wu Y, Huang W, Deng L. Does social trust stimulate university technology transfer? Evidence from China. PLoS One. 2021; 16(8). doi: 10.1371/journal.pone.0256551.

Reference 17: Added article doi.

Li, Y; Tang, YJ. A dynamic capabilities perspective on pro-market reforms and university technology transfer in a transition economy. Technovation.2021;103. doi: 10.1016/j.technovation.2021.102224.

Reference 18: Added authors and magazine abbreviations.

Hou BJ, Hong J, Chen Q, Shi X, Zhou Y. Do academia-industry R&D collaborations necessarily facilitate industrial innovation in China? The role of technology transfer institutions. Eur J Innov Manag. 2019; 22(5):717-746.

Reference 20: Added magazine abbreviations.

Xiong X, Yang GL, Guan ZC. Assessing R&D efficiency using a two-stage dynamic DEA model: A case study of research institutes in the Chinese Academy of Sciences. J Informetr. 2018; 12(3):784-805.

Reference 21: Added authors and magazine abbreviations.

Gao Q, He F, Moosa A, Lv Q. The Efficiency of Technology Transfer in Chinese Key Universities. J Sci Ind Res. 2019; 78(9):582-588.

Reference 22: Added magazine abbreviations.

Son H, Chung Y, Yoon S. Is the alignment between public research organizations’ R&D competence and policies really critical for technology transfer? Sci Public Policy. 2021; 48(1):93-104.

Reference 23: Added magazine abbreviations.

Di F. Transfer Benefit Evaluation on University S&T Achievements Based on Bootstrap-DEA. Educ Sci-Theory Pract. 2018; 18(5):1125-1137.

Reference 26: Added magazine abbreviations.

Lampe HW, Hilgers D. Trajectories of efficiency measurement: A bibliometric analysis of DEA and SFA. Eur J Oper Res. 2015; 240(1):1-21.

Reference 27: Added article specific time of publication.

Li F, Zhang S, Jin YH. Sustainability of University Technology Transfer: Mediating Effect of Inventor's Technology Service. Sustainability. 2018 Sep 13. doi:10.3390/su10062085.

Reference 29: Added authors and magazine abbreviations.

29. Calcagnini G, Favaretto I, Giombini G, Perugini F, Rombaldoni R. The role of universities in the location of innovative start-ups. J Technol Transf. 2016;41(4):670-693.

Reference 30: Added authors and magazine abbreviations.

30. Colombelli A, De Marco A, Paolucci E, Ricci R, Scellato G. University technology transfer and the evolution of regional specialization: the case of Turin. J Technol Transf. 2021; 46(4):933-960.

Reference 31: Added magazine abbreviations.

Boschma, R. Proximity and innovation: A critical assessment. Reg Stud. 2005; 39(1):61-74.

Reference 32: Added magazine abbreviations.

Siegel DS, Waldman DA, Atwater LE, Link AN. Toward a model of the effective transfer of scientific knowledge from academicians to practitioners: qualitative evidence from the commercialization of university technologies. J Eng Technol Manage. 2004; 21(1-2):115-142.

Reference 33: Added magazine abbreviations.

Gregorio DD, Shane S. Why do some universities generate more start-ups than others? Res Policy. 2003;32(2):209-227.

Reference 35: Added magazine abbreviations.

Caldera A, Debande O. Performance of Spanish universities in technology transfer: An empirical analysis. Res Policy. 2010; 39(9):1160-1173.

Reference 36: Added magazine abbreviations, article specific time of publication and doi.

Sun, Chia-Chi. Evaluating the Intertwined Relationships of the Drivers for University Technology Transfer. Appl Sci-Basel. 2021 Nov 25; 20(11). doi: 10.3390/app11209668.

Reference 37: Added authors, magazine abbreviations and article specific time of publication.

Ar IM, Temel S, Dabic M, Howells J, Mert A, Yesilay RB. The Role of Supporting Factors on Patenting Activities in Emerging Entrepreneurial Universities. IEEE Trans Eng Manage. 2021 Dec 28. doi:10.1109/TEM.2021.3069147.

Reference 38: Added authors, magazine abbreviations and Volume.

Jonek-Kowalska I, Musiol-Urbanczyk A, Podgorska M, Wolny M . Does motivation matter in evaluation of research institutions? Evidence from Polish public universities. Technol Soc. 2021; 67. doi: 10.1016/j.techsoc.2021.101782.

Reference 39: Added magazine abbreviations and pages.

Lafuente E, Berbegal-Mirabent J. Assessing the productivity of technology transfer offices: an analysis of the relevance of aspiration performance and portfolio complexity. J Technol Transf. 2019; 44(3): 778-801.

Reference 40: Added magazine abbreviations and Volume.

Wonglimpiyarat, Jarunee. The innovation incubator, university business incubator and technology transfer strategy: The case of Thailand. Technol Soc. 2016;46: 18-27.

Reference 41: Added authors.

Escobar ESO, Berbegal-Mirabent J, Alegre I, Velasco OGD. Researchers’ willingness to engage in knowledge and technology transfer activities: an exploration of the underlying motivations. R&D Management. 2017; 47(5):715-726.

Reference 42: Added article Issue.

Belenzon, S.; Schankerman, M. University Knowledge Transfer: Private Ownership, Incentives, and Local Development Objectives. J Law Econ. 2009; 52(1): 111-144.

Reference 44: Added authors, magazine abbreviations and article specific time of publication.

Vlaisavljevic V, Medina CC, Looy BV. The role of policies and the contribution of cluster agency in the development of biotech open innovation ecosystem. Technol Forecast Soc Chang. 2020 May 14. doi: 10.1016/j.techfore.2020.119987.

Reference 47: Added article specific time of publication.

Zhao JY, Wu GD. Evolution of the Chinese Industry-University-Research Collaborative Innovation System. Complexity, 2017 May 17. doi:10.1155/2017/4215805.

Reference 51: Added magazine abbreviations.

Munari F, Sobrero M, Toschi L. The university as a venture capitalist? Gap funding instruments for technology transfer. Technol Forecast Soc Chang. 2018; 127:70-84.

Reference 52: Added magazine abbreviations.

Villani E, Rasmussen E, Grimaldi R. How intermediary organizations facilitate university-industry technology transfer: A proximity approach. Technol Forecast Soc Chang. 2017;114:86-102.

Reference 53: Added magazine abbreviations.

Al-Tabbaa O, Ankrah S. Social capital to facilitate 'engineered' university-industry collaboration for technology transfer: A dynamic perspective. Technol Forecast Soc Chang. 2016;104:1-15.

Reference 54: Added authors, magazine abbreviations and article specific time of publication.

Temel S, Dabic M, Ar IM, Howells J, Mert A, Yesilay RB. Exploring the relationship between university innovation intermediaries and patenting performance. Technol Soc. 2021 Aug 13. doi: 10.1016/j.techsoc.2021.101665.

Reference 56: Added article specific time of publication.

Fang HN, Yang Q, Wang JM, et al. Coupling Coordination between Technology Transfer in Universities and High-Tech Industries Development in China. Complexity. 2021 Aug 1. Doi:10.1155/2021/1809005.

Reference 57: Added magazine abbreviations.

Dikbas F. A New Two-Dimensional Rank Correlation Coefficient. Water Resour Manag. 2018; 32(5):1539-1553.

Reference 58: Added magazine abbreviations and article specific time of publication.

Stephanou M, Varughese M. Sequential estimation of Spearman rank correlation using Hermite series estimators. J Multivar Anal. 2021 Oct 10, 186, doi: 10.1016/j.jmva.2021.104783.

Reference 59: Added magazine abbreviations and article specific time of publication.

Yuan YY; Han ZL. Structural characteristics and proximity comparison of China's urban innovation cooperation network. PLoS One. 2021 Aug 26. doi: 10.1371/journal.pone.0255443.

Reference 60: Added article specific time of publication.

Tang, S, Zhang JX, Niu FQ. Spatial-Temporal Evolution Characteristics and Countermeasures of Urban Innovation Space Distribution: An Empirical Study Based on Data of Nanjing High-Tech Enterprises. Complexity. 2020 Nov 4. doi: 10.1155/2020/2905482.

Reference 61: Added magazine abbreviations.

Broekel T, Boschma R. Knowledge networks in the Dutch aviation industry: the proximity paradox. J Econ Geogr. 2012;12(2): 409–433.

Reference 62: Added magazine abbreviations.

Xiong JB, Liu YQ, Lin XG. Geographic distance and pH drive bacterial distribution in alkaline lake sediments across Tibetan Plateau. Environ Microbiol. 2012;14(9):2457-2466.

Reference 63: Added article specific time of publication.

Zheng DF, Hao S, Sun CZ. Spatial Correlation and Convergence Analysis of Eco-Efficiency in China. Sustainability. 2019 Jul 9. doi:10.3390/su11092490.

Reference 64: Added magazine abbreviations.

Arbia G. Spatial Econometrics: Statistical Foundations and Applications to Regional Convergence. J Reg Sci. 2007; 47(3):646-648.

Reference 65: Added authors and article specific time of publication.

Chen Y, Shi H, Ma J, Shi V. The Spatial Spillover Effect in Hi-Tech Industries: Empirical Evidence from China. Sustainability. 2020 Apr 14. doi:10.3390/su12041551.

Reference 66: Added magazine abbreviations.

Hsu PH, Tian X, Xu Y. Financial development and innovation: Cross-country evidence. J financ econ. 2014; 112(1):116-135.

Reference 67: Added magazine abbreviations.

Rousseau, P. L., Wachtel, P. What Is Happening to the Impact of Financial Deepening on Economic Growth. Econ Inq. 2011; 49(1):276-288.

Reference 68: This reference have been removed and replace it with relevant current references as follows:

Mensi W, Hammoudeh S, Tiwari AK, Al-Yahyaee KH. Impact of Islamic banking development and major macroeconomic variables on economic growth for Islamic countries: Evidence from panel smooth transition models. Econ Syst. 2020 May 8. doi: 10.1016/j.ecosys.2019.100739.

Reference 70: Added authors, magazine abbreviations and article specific time of publication.

Gao X, Ji Z, Ahmad F, Draz MU. Financial Support and Growth of Township and Village Enterprises in China: Fresh Evidence From Regional Analyses. Sage Open. 2019 Jul 9, 9(2). doi:10.1177/2158244019858444.

Reference 71: Added magazine abbreviations.

Zhang H, Ding D, Ke L. The Effect of R&D Input and Financial Agglomeration on the Growth Private Enterprises: Evidence from Chinese Manufacturing Industry. Emerg Mark Financ Trade. 2019; 55(10):2298-2313.

Reference 72: Added magazine abbreviations.

Karlsson H K, Mnsson K, Hacker S. Revisiting the nexus of the financial development and economic development: new international evidence using a wavelet approach. Empir Econ. 2021; 60(5): 2323-2350.

Reference 73: Added magazine abbreviations.

Demirel P, Kesidou E. Sustainability-oriented capabilities for eco-innovation: Meeting the regulatory, technology, and market demands. Bus Strateg Environ. 2019; 28(5):847-857.

---

## [Editor Report · Decision Letter 3]

23 May 2022

PONE-D-21-32734R3Spatiotemporal characteristics and influencing factors analysis of universities technology transfer level in China: the perspective of innovation ecosystemPLOS ONE

Dear Dr. Wang,

Thank you for submitting your manuscript to PLOS ONE. After careful consideration, we feel that it has merit but does not fully meet PLOS ONE’s publication criteria as it currently stands. Therefore, we invite you to submit a revised version of the manuscript that addresses the points raised during the review process. There are some issues regarding your writing style. PLOS ONE does not copyedit accepted manuscripts, so the language in submitted articles must be clear, correct, and unambiguous. We may reject papers that do not meet these standards. I recommend that authors seek independent editorial help from a professional proofreader, and attach to the next submission round a copy/proof of the professional English revision.

We look forward to receiving your revised manuscript.

Kind regards,

Stefano Ghinoi, Ph.D.

Academic Editor

PLOS ONE
---

## [Author Response · Author response to Decision Letter 3]

4 Jun 2022

Response to Comments

We would like to express my appreciation to you for suggesting how to improve our paper. We hope that the revised manuscript is now suitable for publication.

For Academic Editor’ comments, thank you for your valuable and thoughtful comments. We have sought independent editorial help from a professional proofreader to improve writing style and correct language. For Journal Requirements, we checked the references again and attached the last revision and the results of this check.

Journal Requirements:

Thank you for your advice. According to your suggestion, we have checked reference list and refined the content and format according to the requirements of the journal. We have deleted Reference 54, because of repeat with Reference 52. The specific modifications are as follows.

Reference 1: Added magazine abbreviations and article Issue.

Trippl M, Sinozic T, Smith HL. The role of universities in regional development: Conceptual models and policy institutions in the UK, Sweden and Austria. Eur Plan Stud. 2015; 23(9): 1722-1740.

Reference 3: Added article specific time of publication.

Han J. Technology Commercialization through Sustainable Knowledge Sharing from University-Industry Collaborations, with a Focus on Patent Propensity. Sustainability. 2017 Oct 8. doi: 10.3390/su9101808.

Reference 4: Added magazine abbreviations.

Etzkowitz H, Leydesdorff L. The dynamics of innovation: From national systems and “mode 2” to a triple helix of university–industry–government relations. Res Policy. 2000; 29(2):109-123.

Reference 5: Added magazine abbreviations.

Harman G． Australian university research commercialization: perceptions of technology transfer specialists and science and technology academics．J High Educ Policy Manag. 2010;32(1):69-83．

Reference 6: Added magazine abbreviations and article Issue.

Buenstorf G． Is commercialization good or bad for science? Individual - level evidence from the Max Planck Society．Res Policy. 2009; 38(2):281-292．

Reference 7: Changed article published year.

Muscio A. What drives the university use of technology transfer offices? Evidence from Italy. J Technol Transf. 2009; 35(2): 181-202.

Reference 9: Added magazine abbreviations.

Markman GD, Phan PH, Balkin DB, Gianiodis PT. Entrepreneurship and university-based technology transfer. J Bus Ventur. 2005;20(2):241-263.

Reference 10: Added magazine abbreviations.

Sampat B N. Patenting and US academic research in the 20thcentury: The world before and after Bayh-Dole. Res Policy. 2006;35(6):772-789．

Reference 13: Changed citation format and added article Issue.

Chen, A., Patton, D. & Kenney, M. University technology transfer in China: a literature review and taxonomy. J Technol Transf. 2016; 41(5): 891-929.

Reference 14: Added authors and magazine abbreviations.

Yang W, Yu X, Wang D, Yang JR, Zhang B. Spatio-temporal evolution of technology flows in China: patent licensing networks 2000-2017. J Technol Transf. 2021,46(5):1674-1703.

Reference 15: Added authors and magazine abbreviations.

Yuan CH, Li Y, Vlas CO, Peng MW. Dynamic capabilities, subnational environment, and university technology transfer. Strateg Organ. 2018; 16(1):35-60.

Reference 16: Added magazine abbreviations and article doi.

Wu Y, Huang W, Deng L. Does social trust stimulate university technology transfer? Evidence from China. PLoS One. 2021; 16(8). doi: 10.1371/journal.pone.0256551.

Reference 17: Added article doi.

Li, Y; Tang, YJ. A dynamic capabilities perspective on pro-market reforms and university technology transfer in a transition economy. Technovation.2021;103. doi: 10.1016/j.technovation.2021.102224.

Reference 18: Added authors and magazine abbreviations.

Hou BJ, Hong J, Chen Q, Shi X, Zhou Y. Do academia-industry R&D collaborations necessarily facilitate industrial innovation in China? The role of technology transfer institutions. Eur J Innov Manag. 2019; 22(5):717-746.

Reference 20: Added magazine abbreviations.

Xiong X, Yang GL, Guan ZC. Assessing R&D efficiency using a two-stage dynamic DEA model: A case study of research institutes in the Chinese Academy of Sciences. J Informetr. 2018; 12(3):784-805.

Reference 21: Added authors and magazine abbreviations.

Gao Q, He F, Moosa A, Lv Q. The Efficiency of Technology Transfer in Chinese Key Universities. J Sci Ind Res. 2019; 78(9):582-588.

Reference 22: Added magazine abbreviations.

Son H, Chung Y, Yoon S. Is the alignment between public research organizations’ R&D competence and policies really critical for technology transfer? Sci Public Policy. 2021; 48(1):93-104.

Reference 23: Added magazine abbreviations.

Di F. Transfer Benefit Evaluation on University S&T Achievements Based on Bootstrap-DEA. Educ Sci-Theory Pract. 2018; 18(5):1125-1137.

Reference 26: Added magazine abbreviations.

Lampe HW, Hilgers D. Trajectories of efficiency measurement: A bibliometric analysis of DEA and SFA. Eur J Oper Res. 2015; 240(1):1-21.

Reference 27: Added article specific time of publication.

Li F, Zhang S, Jin YH. Sustainability of University Technology Transfer: Mediating Effect of Inventor's Technology Service. Sustainability. 2018 Sep 13. doi:10.3390/su10062085.

Reference 29: Added authors and magazine abbreviations.

29. Calcagnini G, Favaretto I, Giombini G, Perugini F, Rombaldoni R. The role of universities in the location of innovative start-ups. J Technol Transf. 2016;41(4):670-693.

Reference 30: Added authors and magazine abbreviations.

30. Colombelli A, De Marco A, Paolucci E, Ricci R, Scellato G. University technology transfer and the evolution of regional specialization: the case of Turin. J Technol Transf. 2021; 46(4):933-960.

Reference 31: Added magazine abbreviations.

Boschma, R. Proximity and innovation: A critical assessment. Reg Stud. 2005; 39(1):61-74.

Reference 32: Added magazine abbreviations.

Siegel DS, Waldman DA, Atwater LE, Link AN. Toward a model of the effective transfer of scientific knowledge from academicians to practitioners: qualitative evidence from the commercialization of university technologies. J Eng Technol Manage. 2004; 21(1-2):115-142.

Reference 33: Added magazine abbreviations.

Gregorio DD, Shane S. Why do some universities generate more start-ups than others? Res Policy. 2003;32(2):209-227.

Reference 35: Added magazine abbreviations.

Caldera A, Debande O. Performance of Spanish universities in technology transfer: An empirical analysis. Res Policy. 2010; 39(9):1160-1173.

Reference 36: Added magazine abbreviations, article specific time of publication and doi.

Sun, Chia-Chi. Evaluating the Intertwined Relationships of the Drivers for University Technology Transfer. Appl Sci-Basel. 2021 Nov 25; 20(11). doi: 10.3390/app11209668.

Reference 37: Added authors, magazine abbreviations and article specific time of publication.

Ar IM, Temel S, Dabic M, Howells J, Mert A, Yesilay RB. The Role of Supporting Factors on Patenting Activities in Emerging Entrepreneurial Universities. IEEE Trans Eng Manage. 2021 Dec 28. doi:10.1109/TEM.2021.3069147.

Reference 38: Added authors, magazine abbreviations and Volume.

Jonek-Kowalska I, Musiol-Urbanczyk A, Podgorska M, Wolny M . Does motivation matter in evaluation of research institutions? Evidence from Polish public universities. Technol Soc. 2021; 67. doi: 10.1016/j.techsoc.2021.101782.

Reference 39: Added magazine abbreviations and pages.

Lafuente E, Berbegal-Mirabent J. Assessing the productivity of technology transfer offices: an analysis of the relevance of aspiration performance and portfolio complexity. J Technol Transf. 2019; 44(3): 778-801.

Reference 40: Added magazine abbreviations and Volume.

Wonglimpiyarat, Jarunee. The innovation incubator, university business incubator and technology transfer strategy: The case of Thailand. Technol Soc. 2016;46: 18-27.

Reference 41: Added authors.

Escobar ESO, Berbegal-Mirabent J, Alegre I, Velasco OGD. Researchers’ willingness to engage in knowledge and technology transfer activities: an exploration of the underlying motivations. R&D Management. 2017; 47(5):715-726.

Reference 42: Added article Issue.

Belenzon, S.; Schankerman, M. University Knowledge Transfer: Private Ownership, Incentives, and Local Development Objectives. J Law Econ. 2009; 52(1): 111-144.

Reference 44: Added authors, magazine abbreviations and article specific time of publication.

Vlaisavljevic V, Medina CC, Looy BV. The role of policies and the contribution of cluster agency in the development of biotech open innovation ecosystem. Technol Forecast Soc Chang. 2020 May 14. doi: 10.1016/j.techfore.2020.119987.

Reference 47: Added article specific time of publication.

Zhao JY, Wu GD. Evolution of the Chinese Industry-University-Research Collaborative Innovation System. Complexity, 2017 May 17. doi:10.1155/2017/4215805.

Reference 51: Added magazine abbreviations.

Munari F, Sobrero M, Toschi L. The university as a venture capitalist? Gap funding instruments for technology transfer. Technol Forecast Soc Chang. 2018; 127:70-84.

Reference 52: Added magazine abbreviations.

Villani E, Rasmussen E, Grimaldi R. How intermediary organizations facilitate university-industry technology transfer: A proximity approach. Technol Forecast Soc Chang. 2017;114:86-102.

Reference 53: Added magazine abbreviations.

Al-Tabbaa O, Ankrah S. Social capital to facilitate 'engineered' university-industry collaboration for technology transfer: A dynamic perspective. Technol Forecast Soc Chang. 2016;104:1-15.

Reference 54: Added authors, magazine abbreviations and article specific time of publication.

Temel S, Dabic M, Ar IM, Howells J, Mert A, Yesilay RB. Exploring the relationship between university innovation intermediaries and patenting performance. Technol Soc. 2021 Aug 13. doi: 10.1016/j.techsoc.2021.101665.

Reference 56: Added article specific time of publication.

Fang HN, Yang Q, Wang JM, et al. Coupling Coordination between Technology Transfer in Universities and High-Tech Industries Development in China. Complexity. 2021 Aug 1. Doi:10.1155/2021/1809005.

Reference 57: Added magazine abbreviations.

Dikbas F. A New Two-Dimensional Rank Correlation Coefficient. Water Resour Manag. 2018; 32(5):1539-1553.

Reference 58: Added magazine abbreviations and article specific time of publication.

Stephanou M, Varughese M. Sequential estimation of Spearman rank correlation using Hermite series estimators. J Multivar Anal. 2021 Oct 10, 186, doi: 10.1016/j.jmva.2021.104783.

Reference 59: Added magazine abbreviations and article specific time of publication.

Yuan YY; Han ZL. Structural characteristics and proximity comparison of China's urban innovation cooperation network. PLoS One. 2021 Aug 26. doi: 10.1371/journal.pone.0255443.

Reference 60: Added article specific time of publication.

Tang, S, Zhang JX, Niu FQ. Spatial-Temporal Evolution Characteristics and Countermeasures of Urban Innovation Space Distribution: An Empirical Study Based on Data of Nanjing High-Tech Enterprises. Complexity. 2020 Nov 4. doi: 10.1155/2020/2905482.

Reference 61: Added magazine abbreviations.

Broekel T, Boschma R. Knowledge networks in the Dutch aviation industry: the proximity paradox. J Econ Geogr. 2012;12(2): 409–433.

Reference 62: Added magazine abbreviations.

Xiong JB, Liu YQ, Lin XG. Geographic distance and pH drive bacterial distribution in alkaline lake sediments across Tibetan Plateau. Environ Microbiol. 2012;14(9):2457-2466.

Reference 63: Added article specific time of publication.

Zheng DF, Hao S, Sun CZ. Spatial Correlation and Convergence Analysis of Eco-Efficiency in China. Sustainability. 2019 Jul 9. doi:10.3390/su11092490.

Reference 64: Added magazine abbreviations.

Arbia G. Spatial Econometrics: Statistical Foundations and Applications to Regional Convergence. J Reg Sci. 2007; 47(3):646-648.

Reference 65: Added authors and article specific time of publication.

Chen Y, Shi H, Ma J, Shi V. The Spatial Spillover Effect in Hi-Tech Industries: Empirical Evidence from China. Sustainability. 2020 Apr 14. doi:10.3390/su12041551.

Reference 66: Added magazine abbreviations.

Hsu PH, Tian X, Xu Y. Financial development and innovation: Cross-country evidence. J financ econ. 2014; 112(1):116-135.

Reference 67: Added magazine abbreviations.

Rousseau, P. L., Wachtel, P. What Is Happening to the Impact of Financial Deepening on Economic Growth. Econ Inq. 2011; 49(1):276-288.

Reference 68: This reference have been removed and replace it with relevant current references as follows:

Mensi W, Hammoudeh S, Tiwari AK, Al-Yahyaee KH. Impact of Islamic banking development and major macroeconomic variables on economic growth for Islamic countries: Evidence from panel smooth transition models. Econ Syst. 2020 May 8. doi: 10.1016/j.ecosys.2019.100739.

Reference 70: Added authors, magazine abbreviations and article specific time of publication.

Gao X, Ji Z, Ahmad F, Draz MU. Financial Support and Growth of Township and Village Enterprises in China: Fresh Evidence From Regional Analyses. Sage Open. 2019 Jul 9, 9(2). doi:10.1177/2158244019858444.

Reference 71: Added magazine abbreviations.

Zhang H, Ding D, Ke L. The Effect of R&D Input and Financial Agglomeration on the Growth Private Enterprises: Evidence from Chinese Manufacturing Industry. Emerg Mark Financ Trade. 2019; 55(10):2298-2313.

Reference 72: Added magazine abbreviations.

Karlsson H K, Mnsson K, Hacker S. Revisiting the nexus of the financial development and economic development: new international evidence using a wavelet approach. Empir Econ. 2021; 60(5): 2323-2350.

Reference 73: Added magazine abbreviations.

Demirel P, Kesidou E. Sustainability-oriented capabilities for eco-innovation: Meeting the regulatory, technology, and market demands. Bus Strateg Environ. 2019; 28(5):847-857.

---

## [Editor Report · Decision Letter 4]

12 Jun 2022

Spatiotemporal characteristics and influencing factor analysis of universities’ technology transfer level in China: the perspective of innovation ecosystems

PONE-D-21-32734R4

Dear Dr. Wang,

We’re pleased to inform you that your manuscript has been judged scientifically suitable for publication and will be formally accepted for publication once it meets all outstanding technical requirements.

Kind regards,

Stefano Ghinoi, Ph.D.

Academic Editor

PLOS ONE
---

## [Editor Report · Acceptance letter]

22 Jun 2022

PONE-D-21-32734R4 

Spatiotemporal characteristics and influencing factor analysis of universities’ technology transfer level in China: the perspective of innovation ecosystems 

Dear Dr. Wang:

I'm pleased to inform you that your manuscript has been deemed suitable for publication in PLOS ONE. Congratulations! Your manuscript is now with our production department. 

Kind regards, 

on behalf of

Dr. Stefano Ghinoi 

Academic Editor

PLOS ONE